# Efficient Test-Time Adaptation for Super-Resolution with Second-Order Degradation and Reconstruction

**Zeshuai Deng**[1*]  **Zhuokun Chen**[1 2*]  **Shuaicheng Niu**[1*]  **Thomas H. Li**[5]
**Bohan Zhuang**[3†]  **Mingkui Tan**[1 2 4†]

[1]South China University of Technology, [2]Pazhou Lab, [3]ZIP Lab, Monash University,
[4]Key Laboratory of Big Data and Intelligent Robot, Ministry of Education,
[5]Peking University Shenzhen Graduate School

## Abstract

Image super-resolution (SR) aims to learn a mapping from low-resolution (LR) to high-resolution (HR) using paired HR-LR training images. Conventional SR methods typically gather the paired training data by synthesizing LR images from HR images using a predetermined degradation model, *e.g.*, Bicubic down-sampling. However, the realistic degradation type of test images may mismatch with the training-time degradation type due to the dynamic changes of the real-world scenarios, resulting in inferior-quality SR images. To address this, existing methods attempt to estimate the degradation model and train an image-specific model, which, however, is quite time-consuming and impracticable to handle rapidly changing domain shifts. Moreover, these methods largely concentrate on the estimation of one degradation type (*e.g.*, blur degradation), overlooking other degradation types like noise and JPEG in real-world test-time scenarios, thus limiting their practicality. To tackle these problems, we present an efficient test-time adaptation framework for SR, named SRTTA, which is able to quickly adapt SR models to test domains with different/unknown degradation types. Specifically, we design a second-order degradation scheme to construct paired data based on the degradation type of the test image, which is predicted by a pre-trained degradation classifier. Then, we adapt the SR model by implementing feature-level reconstruction learning from the initial test image to its second-order degraded counterparts, which helps the SR model generate plausible HR images. Extensive experiments are conducted on newly synthesized corrupted DIV2K datasets with 8 different degradations and several real-world datasets, demonstrating that our SRTTA framework achieves an impressive improvement over existing methods with satisfying speed. The source code is available at https://github.com/DengZeshuai/SRTTA.

## 1 Introduction

Image super-resolution (SR) aims to reconstruct plausible high-resolution (HR) images from the given low-resolution (LR) images, which is widely applied in microscopy [42, 44], remote sensing [17, 35] and surveillance [40, 56]. Most previous SR methods [11, 31, 60, 8] hypothesize that the LR images are downsampled from HR images using a predefined degradation model, *e.g.*, Bicubic down-sampling. However, due to the diverse imaging sensors and multiple propagations on the Internet, real-world images may contain different degradation types (*e.g.*, Gaussian blur, Poisson noise, and JPEG artifact) [32, 52, 53, 25]. Besides, the realistic degradations of real-world images

---

*Equal contribution. Email: sedengzeshuai@mail.scut.edu.cn, {caesard216, niushuaicheng}@gmail.com
†Corresponding author. Email: mingkuitan@scut.edu.cn, bohan.zhuang@gmail.com

37th Conference on Neural Information Processing Systems (NeurIPS 2023).

may dynamically change, which are often different from the training one, limiting the performance of pre-trained SR models in dynamically changing test-time environments.

Recently, zero-shot SR methods [43, 9, 12] are proposed to train an image-specific SR model for each test image to alleviate the degradation shift issue. For example, ZSSR [43] uses a predefined/estimated degradation model to generate an image with a lower resolution for each test image. With this downsampled image and the test image, they can train an image-specific SR model to super-resolve the test image. Moreover, DualSR [12] estimates the degradation model and trains the SR model simultaneously to achieve better performance. However, these methods usually require thousands of iterations to estimate the degradation model or train the SR model, which is very time-consuming. Thus, these methods cannot handle real-world test images with rapidly changing domain shifts.

To reduce the inference time of zero-shot methods, some recent works [45, 41] introduce meta-learning [13] to accelerate the adaptation of the SR model, which still requires a predefined/estimated degradation model to construct paired data to update the model. However, most degradation estimation methods [2, 30] focus on the estimation of one degradation type, which limits the adaptation of SR models to test images with other degradations. Recently, test-time adaptation (TTA) methods [46, 47, 57, 38, 48, 39] are proposed to quickly adapt the pre-trained model to the test data in target domain without accessing any source training data. These methods often use simple augmentation operations (*e.g.*, rotation or horizontal flip) on the test image, and construct the pseudo label as the average of the predicted results [57, 48]. For image SR, the pseudo-HR image constructed using this scheme [57, 48] may still contain the degradation close to the test image (*e.g.*, Gaussian blur). With such a pseudo-HR image, the adapted SR model may not be able to learn how to remove the degradation from the test image (see results in Table 1). Therefore, how to quickly and effectively construct the paired data to encourage the SR model to remove the degradation is still an open question.

In this paper, we propose a super-resolution test-time adaptation framework (SRTTA) to adapt a trained super-resolution model to target domains with unknown degradations, as shown in Figure 1. When the degradation shift issue occurs, the key challenge is how to quickly and effectively construct (pseudo) paired data to adapt SR models to the target domain without accessing any clean HR images. To this end, we propose a second-order degradation scheme to construct (pseudo) paired data. Specifically, with a pre-trained degradation classifier, we quickly identify the degradation type from the test images and randomly generate a set of degradations to obtain the second-order degraded images. The paired data, which consists of the second-order degraded images and the test image, enables a rapid adaptation of SR models to the target domain with different degradations. To facilitate the learning of reconstruction, we design a second-order reconstruction loss to adapt the pre-trained model using the paired data in a self-supervised manner. After fast adaptation using our method, the SR model is able to learn how to remove this kind of degradations and generate plausible HR images. Moreover, we also design an adaptive parameter preservation strategy to preserve the knowledge of the pre-trained model to avoid the catastrophic forgetting issue in long-term adaptation. Last but not least, we use eight different degradations to construct two new benchmarks, named DIV2K-C and DIV2K-MC, to comprehensively evaluate the practicality of our method. Experimental results on both our synthesized datasets and several real-world datasets demonstrate that our SRTTA is able to quickly adapt the SR model to the test-time images and achieve an impressive improvement.

Our main contributions are summarized as follows:

- **A novel test-time adaptation framework for image super-resolution**: We propose a super-resolution test-time adaptation (SRTTA) framework to adapt any pre-trained SR models to different target domains during the test time. Without accessing any ground-truth HR images, our SRTTA is applicable to practical scenarios with unknown degradation in a self-supervised manner.

- **A fast data construction scheme with second-order degradation**: We use a pre-trained classifier to identify the degradation type for a test image and construct the paired data using our second-order degradation scheme. Since we do not estimate the parameters of the degradation model, our scheme enables a rapid model adaptation to a wide range of degradation shifts.

- **New test datasets with eight different domains**: We construct new test datasets named DIV2K-C and DIV2K-MC, which contain eight common degradations, to evaluate the practicality of different SR methods. Experimental results on both synthesized datasets and real-world datasets demonstrate the superiority of our SRTTA, *e.g.*, 0.84 dB PSNR improvement on DIV2K-C over ZSSR [43].

## 2  Related Work

**Real-world super-resolution.** To alleviate the domain shift issues, GAN-based methods [54, 5, 24, 36] tend to learn the degradation model of the real-world images in the training stage. These methods often train a generator that explicitly learns the degradation model of real-world images. Besides, some methods [55, 60, 49] try to enumerate most of the degradation models that can be encountered in real-world applications. Based on the estimated/predefined degradation models, these methods can generate LR images whose distribution is similar to real-world images. However, due to the complex and unknown processing of real-world images, it is hard to mimic all types of degradation during the training phase. Instead, some existing methods [15, 2, 22, 21, 34] try to estimate the image-specific degradation model during the test time, which helps to reconstruct more plausible HR images. For instance, optimization-based methods [15, 22, 21] estimate the blur kernel and SR image together in an iterative manner. However, these methods cannot generate satisfactory results when the test images contain different types of degradation (*e.g.*, Poisson noise and JPEG artifact) [32]. Thus, these methods still suffer from the domain shift on test images with unknown degradation.

**Zero-shot super-resolution.** Zero-shot methods [43, 9, 45, 41, 12] aim to train an image-specific SR model for each test image to alleviate the domain shift issue. These methods [43, 12] use a predefined/estimated degradation model to generate an image with a lower resolution from each test image. To estimate the image-specific degradation in a zero-shot manner, KernelGAN [2] utilizes the internal statistics of each test image to learn the degradation model specifically and then uses ZSSR [43] to train an SR model with the estimated degradation. However, these methods usually require a lot of time to estimate the degradation model or train the SR model. MZSR [45] and MLSR [41] are proposed to reduce the number of iteration steps for each test image during test time. Recently, DDNM [51] was proposed to use a pre-trained diffusion model to ensure the generated images obey the distribution of natural images. However, these methods still require a predefined (Bicubic downsampling) or an estimated degradation model. The predefined Bicubic degradation suffers from the domain shift due to its difference from the underlying degradation of real-world images. The estimation methods [2, 30] may focus on the estimation of a single degradation type (*e.g.*, blur) while ignoring other degradation. Thus, these methods often result in unsatisfactory HR images for the test images with different degradation types [32]. In this paper, we use a degradation classifier to quickly recognize the degradation type and randomly generate the degradations with this type. Therefore, we do not need to estimate the degradation model, which is time-saving.

**Test-time adaptation.** Recently, test-time adaptation (TTA) methods [46, 47, 57, 38, 48, 6, 39, 59] have been proposed to alleviate the domain shift issue by online updating the pre-trained model on the test data. TTT [46] uses an auxiliary head to learn the test image information from the self-supervised task. Tent [47] proposed to adapt the pre-trained model with entropy-based loss in an unsupervised manner. CoTTA [48] uses a weight-averaged pseudo-label over training steps to guide the pre-trained model adaptation. However, these methods are mainly developed for image classification and may ignore the characteristics of image super-resolution. Thus, these methods may not be effective in adapting the SR model to remove the degradation from the test image. In this paper, we focus on the image SR task and address the degradation shift issue with our SRTTA framework.

## 3  Preliminary and Problem Definition

**Notation.** Without loss of generality, let $\mathbf{y}$ be a clean high-resolution (HR) image and $\mathbf{x}_c$ be the clean low-resolution (LR) image downsampled from $\mathbf{y}$ using Bicubic interpolation, *i.e.*, $\mathbf{x}_c = \mathbf{y} \downarrow_s$, where $s$ is the scale factor of Bicubic downsampling. Let $\mathbf{x}$ denote a real-world test image degraded from $\mathbf{y}$, *i.e.*, $\mathbf{x} = \mathbf{D}(\mathbf{y})$, where $\mathbf{D}(\cdot)$ is the degradation process. We use $\mathbf{x}_{sd}$ to denote the LR image that is further degraded from the real-world image $\mathbf{x}$. In this paper, we call the test image x as a **first-order** degraded image and the image $\mathbf{x}_{sd}$ degraded from $\mathbf{x}$ as a **second-order** degraded image. $f_\theta(\cdot)$ is a super-resolution (SR) model with parameters $\theta$.

**Image degradation.** The degradation process of real-world test images can be modeled by a classical degradation model $\mathbf{D}(\cdot)$ [33, 49]. Formally, let $\mathbf{k}$ be a blur kernel, $\mathbf{n}$ be an additive noise map and $q$ be the quality factor of $JPEG$ compression, the degraded image $\mathbf{x}$ is defined by

$$\mathbf{x} = \mathbf{D}(\mathbf{y}) = [(\mathbf{y} \otimes \mathbf{k}) \downarrow_s + \mathbf{n}]_{JPEG_q}, \tag{1}$$

where $\otimes$ denotes the convolution operation, $\downarrow_s$ denotes the downsampling with a scale factor of $s$, and $JPEG_q$ denotes the JPEG compression with the quality factor $q$. Similarly, the second-order

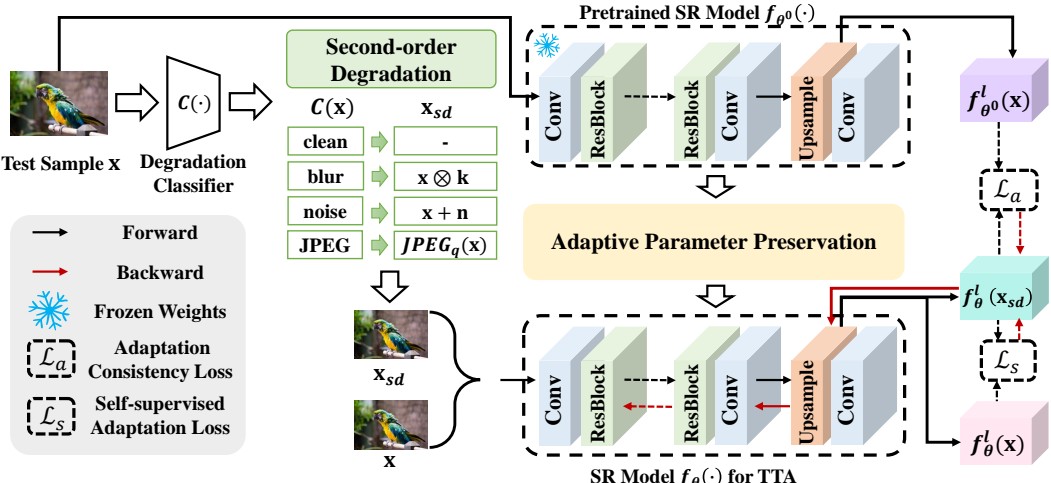

Figure 1: An overall illustration of the proposed super-resolution test-time adaptation (SRTTA) framework. Given a test image $\mathbf{x}$, we use a pre-trained degradation classifier to predict the degradation type $C(\mathbf{x})$, *e.g.*, blur, noise, and JPEG degradation. Based on the predicted degradation type $C(\mathbf{x})$, we construct a set of paired data $\{\mathbf{x}_{sd}^i, \mathbf{x}\}_{i=1}^N$ and adapt the SR model with our adaptation loss $\mathcal{L}_a$ and $\mathcal{L}_s$. When test samples are clean images, we directly use the frozen pre-trained SR model to super-resolve these clean images without adaptation.

degraded image $\mathbf{x}_{sd}$ can be formulated as

$$\mathbf{x}_{sd} = \mathbf{D}(\mathbf{D}(\mathbf{y})) = \mathbf{D}(\mathbf{x}) = [(\mathbf{x} \otimes \mathbf{k}) \downarrow_s + \mathbf{n}]_{JPEG_q}. \tag{2}$$

**Degradation shift between training and testing.** Existing SR methods [31, 60, 16] often construct paired HR-LR training images by either collecting from the real world or synthesizing LR images from HR images via a pre-defined degradation model, *i.e.*, Bicubic down-sampling. However, due to diverse camera sensors and the unknown processing on the Internet, the degradation process of real-world test images may differ from that of training images, *called domain shifts* [27, 48, 32]. In these cases, the SR model often fails to generate satisfactory HR images.

**Motivation and challenges.** Though recently some blind SR methods [12, 43] have been proposed to address the degradation shift issue, they still suffer from two key limitations: low efficiency and narrow focus on a single degradation type, *e.g.*, blur degradation. In this work, we seek to resolve these issues by directly learning from the shifted testing LR image at test time, which poses two major challenges: 1) How to quickly and effectively construct the (pseudo) paired data to adapt SR models to test domains with unknown degradations? and 2) How to design a generalized test-time learning framework that facilitates the removal of various types of degradation, considering that we have only a low-resolution test image at our disposal?

## 4 Efficient Test-Time Adaptation for Image Super-Resolution

In this section, we illustrate our proposed super-resolution test-time adaptation (SRTTA) framework that is able to quickly adapt the pre-trained SR model to real-world images with different degradations. The overall framework and pipeline are shown in Figure 1 and Algorithm 1.

Given a test image $\mathbf{x}$, we first construct the paired data using our second-order degradation scheme. Specifically, we use a pre-trained degradation classifier to recognize the degradation type for the test image. Based on the predicted degradation type, we randomly generate a set of degradation (*e.g.*, a set of blur kernels $\mathbf{k}$) and use them to construct a set of paired data $\{\mathbf{x}_{sd}^i, \mathbf{x}\}_{i=1}^N$. With the paired data, we adapt the pre-trained SR model to remove the degradation from the test image. Notably, before performing the test-time adaptation, we freeze the important parameters to preserve the knowledge of the pre-trained model to alleviate the forgetting problem in long-term adaptation. After adaptation, we use the adapted model to generate the corresponding HR image for the test image.

**Algorithm 1:** The pipeline of the proposed Super-Resolution Test-Time Adaptation.

**Input:** Real-world test images $\{\mathbf{x}_t\}_{t=1}^T$, adaptation iteration steps $S$ for each image, learning rate $\eta$, batch size $N$, preservation ratio $\rho$.

1   Load the pretrained SR models $f_{\theta^0}(\cdot)$ and the degradation classifier $C(\cdot)$.
2   Select and freeze the important parameters using Eqn. (9) with $\rho$.
3   **for** $\mathbf{x}_t$ *in* $\{\mathbf{x}_t\}_{t=1}^T$ **do**
4     **for** $s$ *in* $\{1, 2, ..., S\}$ **do**
5       Construct paired data $\{\mathbf{x}_{sd}^i, \mathbf{x}_t\}_{i=1}^N$ based on $C(\mathbf{x}_t)$ using Eqn. (3);
6       Adapt the SR model using Eqn. (6) with $\eta$;
7     **end**
8   **end**

**Output:** The adapted SR model $f_\theta$, the predictions $\{\hat{\mathbf{y}}_t = f_\theta(\mathbf{x}_t)\}_{t=1}^T$ for all $\mathbf{x}_t$ in $\{\mathbf{x}_t\}_{t=1}^T$.

## 4.1   Adaptive Data Construction with Second-Order Degradation

In this part, we propose a novel second-order degradation scheme to effectively construct paired data, enabling the fast adaptation of SR models to the target domain with different degradations.

Unlike existing methods [43, 2, 30], we consider more degradation types and avoid estimating the degradation model. Existing methods [43, 2, 30] mainly focus on precisely estimating the blur kernels when constructing the lower-resolution images (second-order degraded images), which is time-consuming. Instead, we use a pre-trained degradation classifier to quickly identify the degradation types (blur, noise, and JPEG) of test images, and then we construct the second-order degraded images based on the predicted degradation types. Without the time-consuming degradation estimation process, our scheme enables a fast model adaptation to a wide range of degradation shifts.

**Adaptive data construction.** In this part, we design an adaptive data construction method to obtain the second-order degraded images $\mathbf{x}_{sd}^i$. Specifically, based on the classical degradation model in Eqn. (1), we train a multi-label degradation classifier $C(\cdot)$ to predict the degradation types for each test image, including blur, noise and JPEG degradation types, denoted by $c_b$, $c_n$ and $c_j \in \{0, 1\}$, respectively. With the predicted degradation types, we randomly generate $N$ degradations and construct a set of second-order degraded images $\{\mathbf{x}_{sd}^i\}_{i=1}^N$, which can be formulated as

$$
\begin{aligned}
\mathbf{x}_{sd} &= D(\mathbf{x}, C(\mathbf{x})) = D_j(D_b(\mathbf{x}, c_b) + D_n(c_n), c_j), \\
D_b(\mathbf{x}, c_b) &= c_b(\mathbf{x} \otimes \mathbf{k}) + (1 - c_b)\mathbf{x}, \quad D_n(c_n) = c_n \mathbf{n}, \\
D_j(\mathbf{x}, c_j) &= c_j JPEG_q(\mathbf{x}) + (1 - c_j)\mathbf{x},
\end{aligned}
\tag{3}
$$

where the blur kernel $\mathbf{k}$, noise map $\mathbf{n}$ and quality factor $q$ are randomly generated using a similar recipe of Real-ESRGAN [49]. Unlike previous methods [43, 2], we do not further downsample the test image $\mathbf{x}$ when constructing $\mathbf{x}_{sd}$, since the pretrained SR model has learned the upsampling function (the inverse function of downsampling) during the training phase. Due to the page limit, we put more details in the supplementary materials.

Since the pre-trained SR model has been well-trained on the clean domain (Bicubic downsampling), we simply ignore adapting the clean images in test-time. For these images, we use the pre-trained SR model to super-resolve them, *i.e.*, $\hat{\mathbf{y}} = f_{\theta^0}(\mathbf{x})$ when $c_b = c_n = c_j = 0$.

## 4.2   Adaptation with Second-Order Reconstruction

In our SRTTA framework, we design a self-supervised adaptation loss and an adaptation consistency loss to update the pre-trained SR models to test images with degradation.

**Self-supervised adaptation.** To adapt the pre-trained model to remove the degradation, we design a self-supervised adaptation loss based on the Charbonnier penalty function [4, 28]. Specifically, we encourage the SR model to reconstruct the test images $\mathbf{x}$ from the second-order degraded images $\mathbf{x}_{sd}$ at the feature level, which can be formulated as

$$
\mathcal{L}_s(\mathbf{x}, \mathbf{x}_{sd}) = \sqrt{(f_\theta^l(\mathbf{x}) - f_\theta^l(\mathbf{x}_{sd}))^2 + \epsilon},
\tag{4}
$$

where $f_\theta^l(\cdot)$ denotes the output features of the $l$-th layer. We simply set $f_\theta^l(\cdot)$ to be the output features of the second-to-last convolution layer. $\epsilon$ is a small positive value that is set to $10^{-3}$ empirically.

**Consistency maximization.** To keep the model consistent across adaptation, we design an adaptation consistency loss to encourage the output of the adapted model to be close to that of the pre-trained model, which is formulated as

$$\mathcal{L}_a(\mathbf{x}, \mathbf{x}_{sd}) = \sqrt{(f_{\theta^0}^l(\mathbf{x}) - f_\theta^l(\mathbf{x}_{sd}))^2 + \epsilon}, \tag{5}$$

where $f_{\theta^0}^l(\cdot)$ denotes the output features of the $l$-th layer of the pre-trained SR model.

**Second-order reconstruction loss.** Our final second-order reconstruction loss consists of a self-supervised adaptation loss and an adaptation consistency loss, which is formulated as

$$\mathcal{L} = \mathcal{L}_s(\mathbf{x}, \mathbf{x}_{sd}) + \alpha \mathcal{L}_a(\mathbf{x}, \mathbf{x}_{sd}), \tag{6}$$

where $\alpha$ is a balance hyperparameter (the ablation study can be found in Table 5).

### 4.3 Adaptive Parameter Preservation for Anti-Forgetting

To avoid catastrophic forgetting in long-term adaptation, we propose an adaptive parameter preservation (APP) strategy to freeze the important parameters during adaptation. To select the important parameters, we evaluate the importance of each parameter using the Fisher information matrix [26, 38]. Given a set of collected clean images $\mathcal{D}_c$, we design an augmentation consistency loss $\mathcal{L}_c$ to compute the gradient of each parameter. Based on the gradient, we compute the diagonal Fisher information matrix to evaluate the importance of each parameter $\theta_i^0$, which is formulated as

$$\omega(\theta_i^0) = \frac{1}{|\mathcal{D}_c|} \sum_{\mathbf{x}_c \in \mathcal{D}_c} \left(\frac{\partial \mathcal{L}_c(\mathbf{x}_c)}{\partial \theta_i^0}\right)^2, \tag{7}$$

$$\mathcal{L}_c(\mathbf{x}_c) = \sqrt{(\bar{\mathbf{y}} - f_{\theta^0}(\mathbf{x}_c))^2 + \epsilon}, \ \ s.t. \ \ \bar{\mathbf{y}} = \frac{1}{8} \sum_{i=1}^{8} \mathbf{R}_i(f_{\theta^0}(\mathbf{A}_i(\mathbf{x}_c))), \tag{8}$$

where $\mathbf{A}_i \in \{\mathbf{A}_j\}_{j=1}^8$ is an augmentation operation, which is the random combination of a 90-degree rotation, a horizontal and a vertical flip on the input image $\mathbf{x}_c$. $\mathbf{R}_i$ is the inverse operation of $\mathbf{A}_i$ that rolls back the image $\mathbf{A}_i(\mathbf{x}_c)$ to its original version $\mathbf{x}_c$. With $\omega(\theta_i^0)$, we select the most important parameters using a ratio of $\rho$ and freeze these parameters during the adaptation. The set of selected parameters $\mathcal{S}$ can be formulated as

$$\mathcal{S} = \{\theta_i^0 | \omega(\theta_i^0) > \tau_\rho, \theta_i^0 \in \theta^0\}, \tag{9}$$

where $\tau_\rho$ denotes the first $\rho$-ratio largest value obtained by ranking the value $\omega(\theta_i^0)$, $\rho$ is a hyperparameter to control the ratio of parameters to be frozen. Note that we only need to select the set of significant parameters $\mathcal{S}$ once before performing test-time adaptation.

## 5 Experiments

### 5.1 Experimental Details

**Testing data.** Following ImageNet-C [18], we degraded 100 validation images from the DIV2K [1] dataset into eight domains. We select the eight degradation types that do not extremely change the image content, including Gaussian Blur, Defocus Blur, Glass Blur, Gaussian Noise, Poisson Noise (Shot Noise), Impulse Noise, Speckle Noise, and JPEG compression. In total, we create a new benchmark dataset **DIV2K-C**, which contains 800 images with different single degradation, to evaluate the performance of different SR methods. Besides, we further construct a dataset named **DIV2K-MC**, which consists of four domains with mixed multiple degradations, including BlurNoise, BlurJPEG, NoiseJPEG, and BlurNoiseJPEG. Specifically, test images from the BlurNoiseJPEG domain contain the combined degradation of Gaussian Blur, Gaussian Noise and JPEG simultaneously. Moreover, we also evaluate our SRTTA on real-world images from DPED [23], ADE20K [61] and OST300 [50], whose corresponding ground-truth HR images can not be found. To evaluate the anti-forgetting performance, we use a benchmark dataset Set5 [3].

Table 1: Comparison with existing state-of-the-art SR methods on DIV2K-C for $2\times$ SR regarding PSNR ($\uparrow$) and inference time (second/image), which is measured on a single TITAN XP GPU. The bold number indicates the best result and the underlined number indicates the second best result.

| Method | Gaussian Blur | Defocus Blur | Glass Blur | Gaussian Noise | Poisson Noise | Impulse Noise | Speckle Noise | JPEG | Mean | GPU Time (seconds/image) |
|---|---|---|---|---|---|---|---|---|---|---|
| SwinIR [29] | 30.40 | 25.52 | 27.82 | 25.35 | 22.36 | 15.34 | 30.45 | 30.74 | 26.00 | 13.08 |
| IPT [7] | 28.93 | 24.08 | 26.39 | 22.96 | 20.08 | 13.06 | 28.27 | 28.36 | 24.02 | 55.36 |
| HAT [8] | 29.00 | 24.08 | 26.40 | 22.31 | 19.33 | 11.91 | 28.02 | 28.25 | 23.66 | 25.01 |
| DAN [21] | **34.32** | 25.58 | 31.77 | 26.36 | 23.28 | 11.46 | 30.64 | 31.08 | 26.81 | 3.10 |
| DCLS-SR [34] | 33.93 | 25.55 | **31.98** | 25.45 | 21.59 | 8.12 | 30.66 | 30.86 | 26.02 | 1.45 |
| ZSSR [43] | 29.91 | 25.54 | 27.79 | 26.79 | 24.24 | **19.14** | 30.95 | 31.01 | 26.92 | 117.65 |
| KernalGAN [15]+ZSSR | 30.18 | **25.87** | 29.01 | 21.45 | 19.32 | 17.93 | 25.07 | 26.11 | 24.37 | 231.41 |
| MZSR [45] | 30.14 | 25.54 | 28.03 | 25.94 | 23.48 | 17.05 | 30.00 | 30.49 | 26.33 | 3.34 |
| DualSR [12] | 29.00 | 24.40 | 28.18 | 22.30 | 20.11 | 17.22 | 24.99 | 24.74 | 23.87 | 210.85 |
| DDNM [51] | 28.46 | 24.09 | 26.39 | 24.37 | 21.92 | 13.98 | 28.60 | 28.26 | 24.51 | 2,288.55 |
| EDSR [31] | 30.28 | 25.52 | 27.82 | 25.87 | 22.96 | 15.87 | 30.52 | 30.83 | 26.21 | - |
| TTA-C | 30.21 | 25.50 | 27.79 | 26.37 | 23.57 | 16.41 | 30.25 | 30.91 | 26.38 | 13.59 |
| SRTTA (ours) | 31.07 | 25.86 | 29.01 | **29.65** | 26.69 | 16.15 | **32.33** | **31.30** | **27.76** | 5.38 |
| SRTTA-lifelong (ours) | 31.07 | 25.83 | 29.18 | 29.48 | **27.10** | 16.27 | 31.71 | 31.22 | 27.73 | 5.38 |

**Implementation details and evaluation metric.** We evaluate our approach using the baseline model of EDSR [31] with only 1.37 M parameters for $2 \times$ SR. To demonstrate the effectiveness of our SRTTA, we conduct experiments in two settings, including a **parameter-reset** setting and a **lifelong** setting. In the **parameter-reset** setting, the model parameters will be reset after the adaptation of each domain, which is the **default setting** of our SRTTA. In the **lifelong** setting, the model parameters will never be reset in the long-term adaptation, in this case, we call our methods as SRTTA-lifelong. For the balance weight in Eqn. (6), we set $\alpha$ to 1. For the ratio of parameters to be frozen, we set the $\rho$ to 0.50. Please refer to more details in the supplementary materials.

To compare the inference times of different SR methods, we measure all methods on a TITAN XP with 12G graphics memory for a fair comparison. Due to the memory overload of HAT [8] and DDNM [51], we chop the whole image into smaller patches and process them individually for these two methods. To evaluate the performance of different methods, we use the common metrics PSNR [20] and SSIM [20] and report the results of all methods on the DIV2K-C dataset. Due to the page limit, we mainly report PSNR results and put more results in the supplementary materials.

**Compared methods.** We compare our SRTTA with several state-of-the-art methods including supervised pre-trained SR methods, blind SR methods, zero-shot SR methods, and a TTA baseline method. 1) The supervised pre-trained SR models learn super-resolution knowledge with a predefined degradation process, *i.e.*, the Bicubic downsampling. These methods include SwinIR [29], IPT [7] and HAT [8], and the EDSR baseline [31]. 2) Blind SR models predict the blur kernel of test images and generate HR images simultaneously, *e.g.*, DAN [21] and DCLS-SR [34]. 3) Zero-shot SR models often use a predefined/estimated degradation kernel to construct the LR-HR paired images based on the assumption of the cross-scale patch recurrence [14, 62, 37], and they use the LR-HR paired images to train/update the SR models for each test image. These methods include ZSSR [43], KernelGAN [2]+ZSSR [43], MZSR [9], DualSR [12], DDNM [51]. 4) Moreover, we implement a baseline TTA method (dubbed TTA-C) that utilizes the augmentation consistency loss $\mathcal{L}_c$ in Eqn. (8) to adapt the pre-trained model, similar to MEMO [57] and CoTTA [48].

## 5.2 Comparison with State-of-the-art Methods

We evaluate the effectiveness of our methods in terms of quantitative results and visual results. We report the PSNR results of our SRTTA and existing methods on the DIV2K-C dataset for $2 \times$ SR in Table 1 (more results are put in the supplementary materials). Since DAN [21] and DCLS-SR [34] are trained on paired images with Gaussian Blur degradation, they achieve the best results in Gaussian Blur and Glass Blur degradation. However, they merely achieve a limited performance on average due to the ignoring of the noise and JPEG degradations in their degradation model. Moreover, ZSSR [43] achieves state-of-the-art performance on average due to the thousands of iterations of training steps for each image. Though KernelGAN [15] estimates a more accurate blur kernel and helps to generate more plausible HR images for Gaussian Blur and Glass Blur degradation images, it is harmful to the performance of ZSSR [43] on the noise images, since the degradation model of KernalGAN [15] does not cover the noise degradation. Moreover, the baseline TTA-C may be harmful to adapting the pre-trained model to blur degradation due to the simple augmentation, resulting in a limited

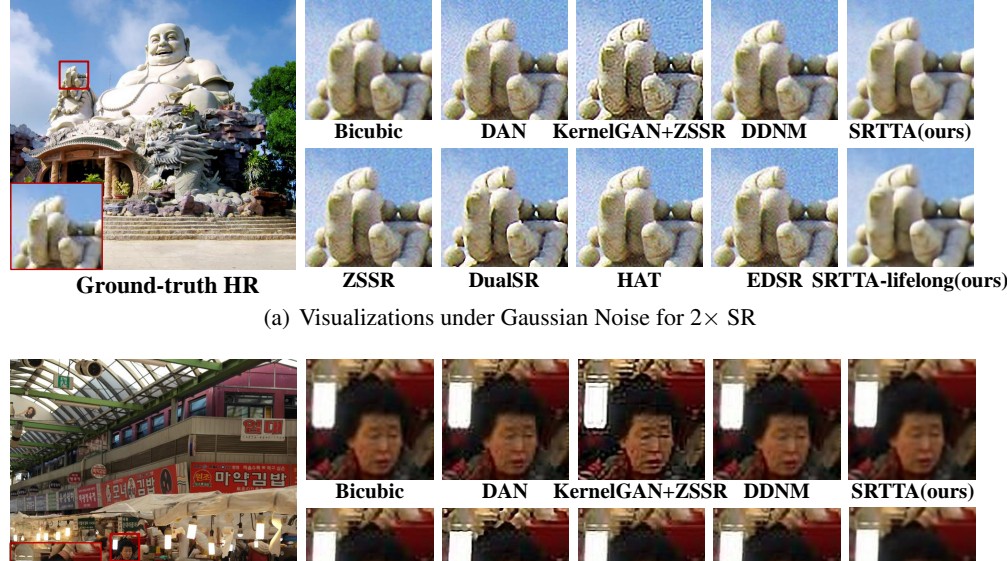

(a) Visualizations under Gaussian Noise for 2× SR

(b) Visualizations under JEPG Compression for 2× SR

Figure 2: Visualization comparison on DIV2K-C test images with degradation for 2× SR.

adaptation performance. Instead, our methods achieve the best performance in terms of PSNR on average. For quality comparison, we provide the visual results of our SRTTA and the compared methods in Figure 2. As shown in Figure 2, our SRTTA is able to remove the degradation and reconstruct clearer SR images with less noise or fewer JPEG artifacts.

**Comparison of inference time.** Moreover, we also compare the inference time of different methods in Table 1. Due to the additional time for test-time adaptation, our SRTTA cannot achieve the best inference speed. However, those methods with less inference time are mostly trained on domains with Gaussian blur degradation only, such as DAN [21], DCLS-SR [34], and MZSR [45], which still suffer from the domain shift issue under noise or JPEG degradation. Instead, with comparable efficiency, our SRTTA achieves an impressive improvement on average for all domains (see results in Table 1). In conclusion, our SRTTA achieves a better tradeoff between performance and efficiency.

**More results on test domains with mixed multiple degradations.** In this part, we evaluate our SRTTA on DIV2K-MC to further investigate the effectiveness of SRTTA under mixed multiple degradations. In Table 2, our SRTTA achieves the best performance on four domains with different mixed degradations, e.g., 26.47 (ZSSR) → 28.48 (our SRTTA-lifelong) regarding the average PSNR metric. These results further validate the effectiveness of our proposed methods.

**Evaluation in terms of human eye-related metrics.** In this part, we further evaluate different methods in terms of the Fréchet Inception Distance (FID) [19] and the Learned Perceptual Image Patch Similarity (LPIPS) distance [58], which correlate well with perceived image quality and are commonly used to evaluate the quality of generated images [10, 55]. We evaluate different methods on two synthesized datasets, including DIV2K-C and DIV2K-MC. As shown in Table 3, our SRTTA achieves the lowest values of both FID and LPIPS scores, demonstrating our SRTTA is able to generate images with higher visual quality.

## 5.3 Further Experiments

In this part, we further conduct several ablation studies to demonstrate the effectiveness of each component of our SRTTA. Last, we evaluate our SRTTA on several datasets with real-world test images and provide the visual comparison results of different methods.

Table 2: Comparison with different SR methods on the synthesized DIV2K-MC dataset. We report the PSNR(↑) values of different methods.

| Methods | Blur Noise | Blur JPEG | Noise JPEG | BlurNoise JPEG | Mean |
|---|---|---|---|---|---|
| SwinIR [29] | 20.91 | 26.83 | 23.86 | 22.77 | 23.59 |
| IPT [7] | 21.28 | 26.83 | 24.15 | 22.96 | 23.81 |
| HAT [8] | 23.41 | 28.86 | 25.69 | 24.42 | 25.59 |
| DAN [21] | 24.14 | 28.95 | 26.20 | 24.82 | 26.03 |
| DCLS-SR [34] | 23.84 | 28.93 | 26.37 | 24.92 | 26.02 |
| ZSSR [43] | 24.9 | 29.02 | 26.68 | 25.24 | 26.47 |
| KernelGAN [2]+ZSSR | 23.08 | 28.32 | 21.90 | 22.76 | 24.02 |
| MZSR [9] | 18.73 | 24.90 | 20.37 | 20.62 | 21.16 |
| DualSR [12] | 25.59 | 28.24 | 23.78 | 24.62 | 25.56 |
| DDNM [51] | 22.62 | 26.82 | 25.11 | 23.81 | 24.59 |
| EDSR [31] | 24.02 | 28.93 | 26.08 | 24.73 | 25.94 |
| TTA-C | 24.29 | 28.93 | 26.35 | 24.91 | 26.12 |
| SRTTA (ours) | 26.93 | 28.93 | **29.13** | 27.12 | 28.02 |
| SRTTA-lifelong (ours) | **27.67** | **29.02** | 29.70 | **27.52** | **28.48** |

Table 3: Comparison with different methods in terms of FID(↓) and LPIPS(↓) on both DIV2K-C and DIV2K-MC datasets.

| Methods | DIV2K-C FID / LPIPS | DIV2K-MC FID / LPIPS |
|---|---|---|
| SwinIR [29] | 72.90 / 0.2441 | 60.62 / 0.2781 |
| IPT [7] | 68.22 / 0.2345 | 58.24 / 0.2453 |
| HAT [8] | 64.92 / 0.2352 | 60.73 / 0.2640 |
| DAN [21] | 73.59 / 0.2260 | 56.96 / 0.2263 |
| DCLS-SR [34] | 83.44 / 0.2472 | 57.93 / 0.2299 |
| ZSSR [43] | 56.66 / 0.1931 | 52.78 / 0.2152 |
| KernelGAN [2]+ZSSR | 88.28 / 0.2160 | 80.19 / 0.2371 |
| MZSR [9] | 68.27 / 0.2085 | 162.72 / 0.4463 |
| DDNM [51] | 70.80 / 0.2101 | 59.64 / 0.2083 |
| EDSR [31] | 69.70 / 0.2242 | 57.95 / 0.2338 |
| TTA-C | 66.95 / 0.2188 | 56.32 / 0.2293 |
| SRTTA (ours) | 54.37 / 0.1877 | 36.88 / 0.1915 |
| SRTTA-lifelong (ours) | **53.30 / 0.1828** | **35.72 / 0.1832** |

Table 4: Effectiveness of components in SRTTA on DIV2K-C.

| Method | Avg. PSNR |
|---|---|
| SRTTA (ours) | **27.76** |
| - w/o classifier $C(\cdot)$ | 26.06 |
| - w/o $\mathcal{L}_s$ | 27.15 |
| - w/o $\mathcal{L}_a$ | 10.24 |

Table 5: Effects of different $\alpha$ (in Eqn. (6)) under parameter-reset setting. We report average PSNR (↑) on DIV2K-C (with degradation shift) and Set5 (w/o degradation shift).

| $\alpha$ | 0 | 0.1 | 0.5 | 1 | 2 | 5 |
|---|---|---|---|---|---|---|
| DIV2K-C | 10.24 | 13.96 | 22.63 | **27.76** | 27.52 | 27.31 |
| Set5 | 11.42 | 37.66 | 37.75 | 34.59 | 35.41 | 35.89 |

**Effect of the degradation classifier $C(\cdot)$.** To investigate the effect of the degradation classifier, we compare our SRTTA with a baseline that does not use the degradation classifier. Specifically, this baseline generates the second-order degraded images with the random degradation type. In this case, a test image with blur degradation can be randomly degraded with blur, noise, or JPEG degradation. We report the PSRN results of our SRTTA and this baseline in Table 4. Experimental results demonstrate the necessity of the degradation classifier in our second-order degradation scheme.

**Effect of $\mathcal{L}_s$ and $\mathcal{L}_a$ in Eqn. (6).** To investigate the effect of the self-supervised adaptation loss $\mathcal{L}_s$ and adaptation consistency loss $\mathcal{L}_a$ in Eqn. (6), we report the mean PSNR results of our SRTTA with $\mathcal{L}_s$-only and $\mathcal{L}_a$-only. As shown in Table 4, without the adaptation consistency loss $\mathcal{L}_a$, the SR models with only the $\mathcal{L}_s$ will inevitably result in a model collapse. This is because the SR model is prone to output the same output for any input images without meaning. When we remove the $\mathcal{L}_s$ loss, SRTTA can only achieve a limited performance, which demonstrates that $\mathcal{L}_s$ truly helps to encourage the SR models to learn how to remove the degradation during the adaptation process. These experimental results demonstrate the effectiveness of the $\mathcal{L}_s$ and $\mathcal{L}_a$ in our framework.

**Effect of the hyper-parameters $\alpha$ in Eqn. (6).** To investigate the effect of the weight of adaptation consistency loss $\alpha$ in Eqn. (6), we report the mean PSNR results of our SRTTA with different $\alpha$ in Table 5. When the $\alpha$ is too small ($\alpha < 1$), the self-supervised degradation loss $\mathcal{L}_s$ dominates the adaptation process and often results in the collapse of SR models. When the $\alpha$ is too large ($\alpha > 1$), the adaptation consistency loss $\mathcal{L}_a$ may have a great constraint on the adapted SR model to be the same as the pre-trained model, which may be harmful to the adaptation of the SR performance. With $\alpha = 1$, our SRTTA achieves the best results on the DIV2K-C dataset on average. Therefore, we simply set the $\alpha$ to be 1 for all other experiments.

**Effect of adaptive parameter preservation in Eqn. (9).** In this part, we investigate the effect of our adaptive parameter preservation (APP) strategy on test-time adaptation. We compare our APP with the existing anti-forgetting method Stochastic Restoration (STO) [48], which randomly restores a different set of parameters (1% parameters) after each adaptation step. Moreover, we also compare our APP with the random selection (RS) baseline, which randomly selects a fixed set of parameters to freeze before adaptation. As shown in Table 6, though STO achieves the best anti-forgetting performance on the clean Set5 dataset, the STO severely hinders the TTA performance. The random selection baseline only achieves a limited performance of both TTA and anti-forgetting. Instead, our APP consistently outperforms the random selection baseline with the same ratio of frozen parameters. Moreover, our APP with $\rho = 0.5$ achieves the best adaptation performance on DIV2K-C (see more

Table 6: Ablation studies of adaptive parameter preservation (APP) strategy on DIV2K-C and Set5 under the lifelong setting. We report PSNR (↑) and results on DIV2K-C are averaged over 8 different degradation types. The compared stochastic restoration (STO) [48] select 1% parameters for each adaptation iteration and Random Selection (RS) is evaluated by selecting different $\rho$ parameters.

| Dataset | STO [48] | RS with different $\rho$ | | | APP with different $\rho$ (ours) | | |
|---|---|---|---|---|---|---|---|
| | | 0.3 | 0.5 | 0.7 | 0.3 | 0.5 | 0.7 |
| DIV2K-C (with degradation shift) | 27.17 | 27.52 | 27.62 | 27.68 | 27.72 | **27.73** | 27.73 |
| Set5 (w/o degradation shift) | 35.57 | 33.95 | 34.02 | 34.24 | 34.11 | 34.23 | 34.38 |

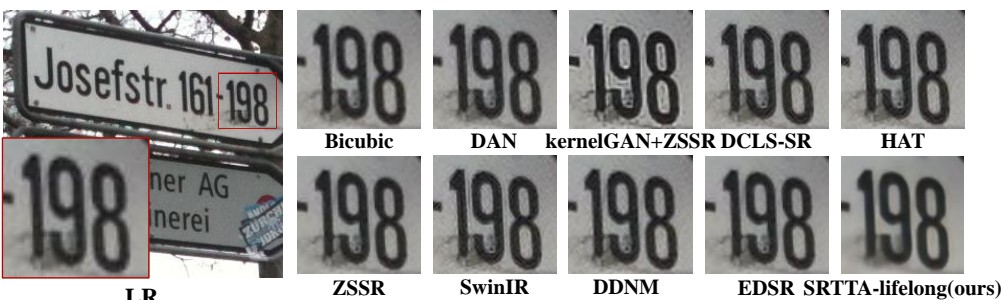

(a) Visualization results of the image from DPED [23].

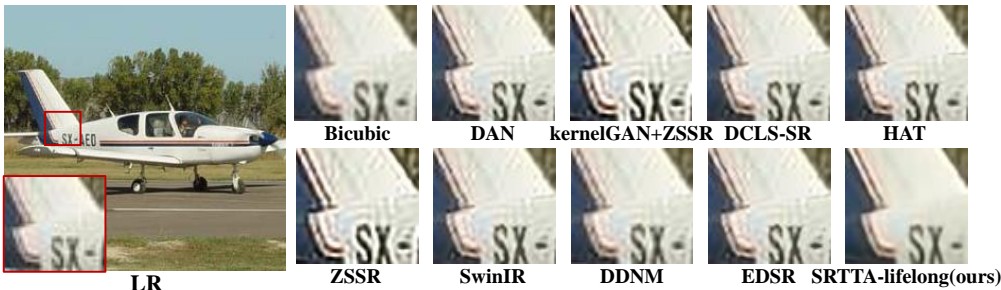

(b) Visualization results of the image from ADE20K [61].

Figure 3: Visualization comparison on real-world test images from DPED [23] and ADE20K [61].

results in the supplementary materials), thus we set the ratio of preservation parameters to be 0.5 in our SRTTA by default. These results demonstrate the effectiveness of our APP strategy.

**Visualization results on the real-world images.** We also provide the visual results of different methods on the real-world images from DPED [23] and ADE20K [61]. We use our SRTTA-lifelong model that has been adapted on DIV2K-C to perform test-time adaptation on the real-world images from DPED [23] and ADE20K [61], respectively. As shown in Figure 3, SRTTA-lifelong is able to generate HR images with fewer artifacts. These results demonstrate that our method is able to be applied to real-world applications. Please refer to more results in the supplementary materials.

## 6 Conclusion

In this paper, we propose a super-resolution test-time adaptation (SRTTA) framework to quickly alleviate the degradation shift issue for image super-resolution (SR). Specifically, we propose a second-order degradation scheme to construct paired data for each test image. Then, our second-order reconstruction loss is able to quickly adapt the pre-trained SR model to the test domains with unknown degradation. To evaluate the effectiveness of our SRTTA, we use eight different types of degradations to synthesize two new datasets named DIV2K-C and DIV2K-MC. Experiments on the synthesized datasets and several real-world datasets demonstrate that our SRTTA is able to quickly adapt the SR model for each image and generate plausible high-resolution images.

## Acknowledgements

This work was partially supported by the National Natural Science Foundation of China (NSFC) (62072190), National Natural Science Foundation of China (NSFC) 61836003 (key project), Program for Guangdong Introducing Innovative and Enterpreneurial Teams 2017ZT07X183, and CCF-Tencent Open Research Fund (CCF-Tencent RAGR20220108).

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

# Supplementary Materials for
# " Efficient Test-Time Adaptation for Super-Resolution with Second-Order Degradation and Reconstruction "

## Contents

# A More Details of the Second-Order Degradation

In our second-order degradation scheme, we randomly generate different degradations to degrade the test image into its second-order degraded counterparts. In this section, we illustrate how to randomly generate different types of degradations. Notably, we degrade the test images on the GPU device to accelerate the degradation process.

## A.1 Random Blur Degradation

Following [26, 22], we model blur degradation as a convolution with a linear Gaussian blur filter/kernel. Given a test image, we randomly generate a set of isotropic or anisotropic Gaussian kernels $\mathbf{k}$ and use them to perform blur degradation on the test image. The probabilities of generating an isotropic kernel and an anisotropic kernel are set to 0.5 and 0.5, respectively. The size of each generated kernel is uniformly sampled from $\{7 \times 7, 9 \times 9, ..., 21 \times 21\}$. We sample the standard deviation of the blur kernel along the two principal axes $\sigma_1$ and $\sigma_2$ uniformly from $[0.2, 3]$. If $\mathbf{k}$ is an isotropic Gaussian blur kernel, we set $\sigma_2$ equal to $\sigma_1$. If $\mathbf{k}$ is an anisotropic Gaussian kernel, we further sample a rotation angle $a$ uniformly from $[-\pi, \pi]$, and use a rotation matrix to transform the generated kernel based on the angle $a$. More details about how to generate a Gaussian blur kernel can refer to Real-ESRGAN [22].

## A.2 Random Noise Degradation

In our second-order degradation, we randomly generate a set of Gaussian noise maps $\mathbf{n}$, and add them to the test image to obtain a set of second-order images with different additive Gaussian noise. With a probability of $60\%$, we generate noise for each channel of RGB images independently, otherwise, we generate the same noise map for all three channels. We first generate a noise map whose values are randomly generated from a normal Gaussian distribution. Then we sample a scale value to enlarge the noise uniformly from $[1, 30]$. More details can be referred to Real-ESRGAN [22].

## A.3 Random JPEG Degradation

For JPEG compression $JPEG_q$, we sample a quality factor $q$ uniformly from $[30, 95]$, and use the JPEG compression with the degradation $q$ to degrade test images into a set of second-order degraded images with compression artifacts. Note that JPEG compression with a lower $q$ compress the test image with a higher compression ratio and the compressed images are generally of a lower quality. To accelerate the degradation process, we use DiffJPEG[1], which is the PyTorch implementation of JPEG compression, to process the test image on the GPU device.

# B The Difference from Real-ESRGAN

In this part, we would like to discuss the difference between our SRTTA with Real-ESRGAN[22], which proposes the concept of second-order degradations, to highlight the contribution of our SRTTA.

## B.1 Solving Different Problems

Real-ESRGAN tries to enumerate all the degradations in real-world scenes and train an SR model to solve the image restoration on any degradation. However, it is non-trivial to obtain all real-world degradations, leading to domain shift issues when encountering unknown degradations during testing, as shown in Figure 11 of real-ESRGAN [22]. Unlike real-ESRGAN [22], our SRTTA aims to adapt the SR models to the test domains when test images contain unknown degradations. Our second-order degradation scheme aims to quickly construct the pseudo-paired data (instead of the paired training data) to adapt the SR model to the test domains.

## B.2 Different Construction Schemes

Real-ESRGAN [22] proposes the second-order degradation to construct the paired training data, whose low-resolution (LR) images are obtained from the ground-truth high-resolution (HR) images.

---

[1] https://github.com/mlomnitz/DiffJPEG

Then, the paired data is used to train an SR model during the **training** phase in a **supervised** learning manner. Notably, the trained Real-ESRGAN [22] model is fixed during the test time. Instead, our second-order degradation scheme constructs the pseudo-paired data using the test images with unknown degradation (first-order degraded images). Our SRTTA model is continuously adapted to different domains during **testing** in a **self-supervised** learning manner.

## C  Experimental Datasets

### C.1  Construction Details of the DIV2K-C Dataset

To evaluate the practicality, we construct a new benchmark dataset named DIV2K-C, which contains eight different degradations. We select the eight degradation types from the 15 corruptions of ImageNet-C [11] that do not extremely change the image content, including Gaussian Blur, Defocus Blur, Glass Blur, Gaussian Noise, Poisson Noise (Shot Noise), Impulse Noise, Speckle Noise, and JPEG compression. Unlike the ImageNet-C [11], we do not use the same degradation level to degrade all test images. Instead, we randomly generate the degradation level and further generate a degradation for each image based on the degradation level. Unlike prior SR methods that investigate a limited number of degradation types, the degradation scenarios we considered are more complex (eight degradation types in total), which is more practical for real-world applications.

Given a degradation, we use the classical image degradation model [17, 22] to generate the low-resolution (LR) test images from the high-resolution clean images. For blur degradation, we perform the blur convolution on the HR images and then use the Bicubic downsampling to obtain the test LR images. For noise and JPEG degradation, we first use the Bicubic downsampling to obtain clean LR images and then perform noise or JPEG degradation on the clean LR images to obtain the final test images. We show the visualization of some examples regarding each degradation type in Figure A.

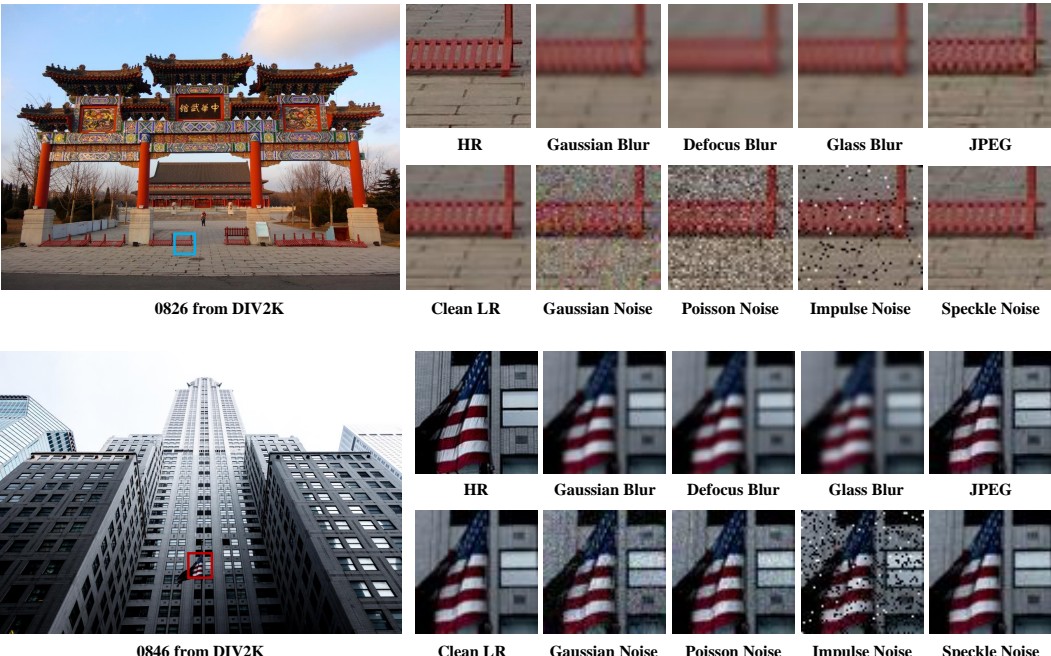

Figure A: The visualization of examples regarding each degradation type on the DIV2K-C dataset.

**Gaussian blur.** Following BSRGAN [26], we generate low-resolution (LR) images with Gaussian blur degradation. We randomly generate an isotropic Gaussian kernel or an anisotropic Gaussian kernel for each high-resolution (HR) image. Then, we use the blur kernel to perform blur convolution on the HR image and use Bicubic downsampling to obtain the final LR test images. For simplicity, we follow the recipe of BSRGAN [26] to generate test images with blur degradation.

**Defocus blur.** To better compare the performance of different SR methods, we also use the common Defocus blur degradation to degrade the HR images and obtain the LR images using the Bicubic downsampling. As illustrated in ImageNet-C [11], Defocus blur often occurs when an image is out of focus when we take pictures. We generate the blur kernel as ImageNet-C [11] and perform a blur convolution on the HR images. But unlike ImageNet-C [11], the degradation level of Defocus blur is randomly sampled from a given range.

**Glass blur.** We also choose another common degradation type, Glass Blur, which appears with "frosted glass" windows or panels [11]. This blur degradation requires two Gaussian blur operations and an operation that locally shuffles pixels between two blur operations. As mentioned above, the degradation level is randomly sampled from a given range, such as the standard deviation of the Gaussian blur kernel or the window size of the shuffling operation.

**Gaussian noise.** To generate test images with Gaussian noise, we sample the noise for each pixel from a normal Gaussian distribution. The mean of the Gaussian distribution is zero, and the standard deviation is uniformly sampled from the range of $\{2/255, 3/255, ..., 25/255\}$. More details can refer to the implementation of BSRGAN [26].

**Poisson (Shot) noise.** Poisson noise, also called Shot noise, can model the sensor noise caused by statistical quantum fluctuations. We randomly generate the Poisson noise map from a Poisson distribution, which has an intensity proportional to the image intensity. Then, we add the generated Poisson noise into the clean LR images to obtain the test images with Poisson noise. More details can refer to the implementation of Real-ESRGAN [22].

**Impulse noise.** Impulse noise is caused by errors in the data transmission generated in noisy sensors or communication channels, or by errors during the data capture from digital cameras [19]. The most common form of Impulse noise is called salt-and-pepper noise. To generate test images with Impulse noise, we uniformly select a set of pixels and replace them with zero or the maximum value (255). More details can be referred to ImageNet-C [11].

**Speckle noise.** Speckle noise is an additive noise where the noise added to a pixel tends to be larger if the original pixel intensity is larger. We first sampled the noise for each pixel from a Gaussian distribution and multiple the noise value by the original pixel. Last, we add the generated noise map into the LR clean images to obtain the final test images with Speckle noise.

**JPEG compression.** For JPEG compression, we use the OpenCV implementation of JPEG compression[2] to degrade the clean LR image into final test images. The compression quality factor $q$ is randomly sampled from $[30, 90]$. We first encode the clean LR images into the bit stream using JPEG compression with the quality factor $q$ and decode the bit stream to obtain the final test images. Note that JPEG compression is a lossy compression technique, so the final test images are inevitably corrupted with JPEG compression artifacts.

### C.2 Construction Details of the DIV2K-MC Dataset

Since the real-world test images may contain multiple degradation types simultaneously, we further develop a new benchmark dataset named DIV2K-MC, which includes four test domains with mixed multiple degradations. The four domains are BlurNoise, BlurJPEG, NoiseJPEG and BlurNoiseJPEG. The test images in the BlurNoiseJPEG domain contain the combined degradation of Gaussian blur, Gaussian noise and JPEG degradations simultaneously.

**BlurNoise.** We generate LR images from HR images using Gaussian blur and Gaussian noise degradation. We first randomly generate a Gaussian blur kernel to perform blur convolution on the HR image. Then, we downsample the resulting image using Bicubic interpolation. Last, we randomly sample a Gaussian noise map and add it to the downsampled image to obtain the final LR image.

**BlurJPEG.** We generate LR images from HR images using Gaussian blur and JPEG degradation. We first randomly generate a Gaussian blur kernel to perform blur convolution on the HR image. Then, the resulting image is downsampled by using Bicubic interpolation. Last, we use JPEG compression with a random quality factor $q$ to compress the downsampled image to obtain the final LR image.

**NoiseJPEG.** We generate LR images from HR images using Gaussian noise and JPEG degradation. We first downsample HR image using Bicubic interpolation. Then, we randomly sample a Gaussian

---

[2]https://github.com/opencv/opencv

noise map and add it to the downsampled image. Last, we use JPEG compression with a random quality factor $q$ to compress the downsampled image to obtain the final LR image.

**BlurNoiseJPEG.** We generate LR images from HR images using Gaussian blur, Gaussian noise and JPEG degradation. We first randomly generate a Gaussian blur kernel to perform blur convolution on the HR image. Second, the resulting image is downsampled by using Bicubic interpolation. Then, we randomly sample a Gaussian noise map and add it to the downsampled image. Last, we use JPEG compression with a random quality factor $q$ to compress the image to obtain the final LR image.

### C.3 More Test Datasets for Test-Time Image Super-Resolution

Moreover, we also evaluate the performance of SR methods on real-world test images from DPED [13], ADE20K [28] and OST300 [23], whose corresponding ground-truth HR images can not be found. To evaluate the anti-forgetting performance, we report the adapted model performance on a clean benchmark dataset Set5 [2] whose images are clean images that are downsampled from HR images with Bicubic interpolation. Thus, these LR images do not contain any degradation.

## D  Implementation Details

### D.1  Implementation Details of the Degradation Classifier

In our second-order degradation scheme, we use a pre-trained degradation classifier to predict the degradation type for each test image. To obtain the pre-trained degradation classifier, we use ResNet-50 [10] as the classifier and train it to recognize the degradation from test images.

In real-world scenes, test images may contain degradations other than these eight degradation types, such as ringing or overshoot artifacts [22], which may be viewed as variations of blur, noise or JPEG. Since it is infeasible to cover all the degradation types in real-world scenes, we make the degradation classifier to predict the coarse-level four classes, including clean, blur, noise and JPEG.

**Training details.** Specifically, we use the 800 training HR images of DIV2K and randomly crop them into patches with the size of $224 \times 224$ (instead of resizing them into $224 \times 224$). Similar to the construction of DIV2K-C, we degrade each patch using a random selection of one of eight degradation types. As for clean data, we do not perform any degradation on the patches. For training, we apply Adam with $\beta_1 = 0.9$, $\beta_2 = 0.999$ and set the batch size as 256. The learning rate is initialized to $10^{-3}$ and decreased to $10^{-6}$ with a cosine annealing out of 400 epochs in total.

**Testing details.** During testing, we directly input the whole test image **with original resolution** into the classifier and output the predicted results to recognize the degradation type. The predicted results of the multi-label degradation classifier $C(\cdot)$ are the probabilities of the three degradations, including blur, noise and JPEG degradation. If the predicted probability of one degradation type is larger than the threshold of 0.5, the test image is considered to contain the degradation of this type. The clean image means that this image does not contain any degradation such as blur, noise, or JPEG, and we directly use the pre-trained SR model to super-resolve these clean test images.

### D.2  More Details of Super-Resolution Test-Time Adaptation

We use the baseline model of EDSR [16] with less than 2M parameters as our pre-trained SR model for $2\times$ and $4\times$ SR. During adaptation, we only update the parameters in the Resblock of the EDSR model. To avoid anti-forgetting, we use five clean test LR images from Set5 [2] to select important parameters to be frozen in Eqn. (9). Moreover, when evaluating the anti-performance Set5 [2], we directly use the adapted model to super-resolve the test images without using the classifier.

In our experiment, we conduct experiments in **parameter-reset** and **lifelong** settings. In the parameter-reset setting, the model parameters will be reset after the adaptation on each domain, which is the default setting of our SRTTA. In the lifelong setting, the model parameters will never be reset in the long-term adaptation, in this case, we call our methods as SRTTA-lifelong.

For test-time adaptation, we use the Adam optimizer with the learning rate of $5 \times 10^{-5}$ for the pre-trained SR models. We set the batch size $N$ to 32, and we randomly crop the test image into $N$ patches of size $96 \times 96$ and $64 \times 64$ for $2\times$ and $4\times$ SR, and degrade them into second-order degraded patches. We perform $S = 10$ iterations of adaptation for each test image. For the balance weight in

 For the ratio of parameters to be frozen, we set the $\rho$ to 0.50. To compare the inference times of different SR methods, we measure all methods on a TITAN XP with 12G graphics memory for a fair comparison. Due to the memory overload of HAT [6] and DDNM [24], we chop the whole image into smaller patches and process them individually for these two methods. In our experiments, we use the bold number to indicate the best result and the underlined number to indicate the second-best result.

# E    More Experimental Results

## E.1    More Results of Test-Time Image Super-Resolution

In this part, we compare our SRTTA with existing SR methods, including supervised pre-trained SR methods, blind SR methods, zero-shot SR methods, and a TTA baseline with consistency loss. 1) The supervised pre-trained SR methods learn to process test images with a predefined degradation process, i.e., the Bicubic downsampling. These methods include the EDSR baseline [16], SwinIR [15], IPT [5] and HAT [6], SRDiff [14], and Real-ESRNet [22]. Note that Real-ESRNet [22] is the PSNR-oriented model of Real-ESRGAN [22], which use a complex combination of different degradation to construct pair training data, including Gaussian blur, Gaussian Noise, Poisson Noise and JPEG compression, and so on. 2) Blind SR models predict the blur kernel of test images and generate HR images simultaneously, we compare with the state-of-the-art DAN [12] and DCLS-SR [18] methods. 3) Zero-shot SR models construct the LR-HR paired images based on the assumption of the cross-scale patch recurrence and train/update their SR model for each test image. These methods include ZSSR [20], KernelGAN [1]+ZSSR [20], MZSR [7], DualSR [8], DDNM [24]. 4) Moreover, we implement a baseline TTA method (dubbed TTA-C) that utilizes the augmentation consistency loss in Eqn. (8) to adapt the pre-trained model, similar to MEMO [27] and CoTTA [21].

We provide more results of our SRTTA for $2\times$ and $4\times$ SR in terms of PSNR and SSIM metrics. As shown in Table A and Table B, our SRTTA consistently outperforms existing SR methods on the DIV2K-C dataset on average. We further provide more visualization results in Figure D, E, F and G, which demonstrate our SRTTA is able to remove the degradation from test images and generate plausible HR images. These results demonstrate that our SRTTA is able to quickly adapt the pre-trained SR model to the test images with different degradation.

Table A: We report the PSNR/SSIM results of all corruption fields in DIV2K-C for $2\times$ SR.

| Methods | GaussianBlur | DefocusBlur | GlassBlur | GaussianNoise | PossionNoise | ImpulseNoise | SpeckleNoise | JPEG | Mean | GPU Time |
|---|---|---|---|---|---|---|---|---|---|---|
| | PSNR/SSIM | PSNR/SSIM | PSNR/SSIM | PSNR/SSIM | PSNR/SSIM | PSNR/SSIM | PSNR/SSIM | PSNR/SSIM | PSNR/SSIM | (seconds/image) |
| Bicubic | 28.04/0.803 | 24.10/**0.784** | 26.31/0.745 | 25.35/0.554 | 23.33/0.496 | 15.28/0.324 | 28.65/0.774 | 28.28/0.806 | 24.92/0.661 | - |
| SwinIR [15] | 30.40/0.838 | 25.52/0.673 | 27.82/0.773 | 25.35/0.510 | 22.36/0.428 | 15.34/0.242 | 30.45/0.774 | 30.74/0.846 | 26.00/0.636 | 13.08 |
| IPT [5] | 28.93/0.820 | 24.08/0.640 | 26.39/0.749 | 22.96/0.439 | 20.08/0.369 | 13.06/0.241 | 28.27/0.728 | 28.36/0.804 | 24.02/0.599 | 55.36 |
| HAT [6] | 29.00/0.821 | 24.08/0.640 | 26.40/0.749 | 22.31/0.417 | 19.33/0.349 | 11.91/0.192 | 28.02/0.722 | 28.25/0.802 | 23.66/0.587 | 25.01 |
| DAN [12] | **34.32/0.916** | 25.58/0.673 | 31.77/0.872 | 26.36/0.558 | 23.28/0.461 | 11.46/0.203 | 30.64/0.777 | 31.08/0.857 | 26.81/0.665 | 3.10 |
| DCLS-SR [18] | 33.93/0.914 | 25.55/0.671 | **31.98/0.872** | 25.45/0.521 | 21.59/0.415 | 8.12/0.112 | 30.66/0.784 | 30.86/0.848 | 26.02/0.642 | 1.45 |
| ZSSR [20] | 29.91/0.831 | 25.54/0.674 | 27.79/0.771 | 26.79/0.590 | 24.24/0.509 | **19.14/0.375** | 30.95/0.813 | 31.01/0.853 | 26.92/0.677 | 117.65 |
| KernalGAN [1]+ZSSR | 30.18/0.859 | **25.87**/0.679 | 29.01/0.808 | 21.45/0.436 | 19.32/0.366 | 17.93/0.354 | 25.07/0.686 | 26.11/0.774 | 24.37/0.620 | 231.41 |
| MZSR [7] | 30.14/0.838 | 25.54/0.670 | 28.03/0.777 | 25.94/0.543 | 23.48/0.472 | 17.05/0.314 | 30.00/0.771 | 30.49/0.845 | 26.33/0.654 | 3.34 |
| DualSR [8] | 29.00/0.854 | 24.40/0.640 | 28.18/0.805 | 22.30/0.509 | 20.11/0.436 | 17.22/0.376 | 24.99/0.738 | 24.74/0.751 | 23.87/0.639 | 210.85 |
| DDNM [24] | 28.46/0.808 | 24.09/0.636 | 26.39/0.744 | 24.37/0.497 | 21.92/0.432 | 13.98/0.310 | 28.60/0.753 | 28.26/0.802 | 24.51/0.623 | 2,288.55 |
| EDSR [16] | 30.28/0.837 | 25.52/0.673 | 27.82/0.773 | 25.87/0.536 | 22.96/0.449 | 15.87/0.269 | 30.52/0.778 | 30.83/0.847 | 26.21/0.645 | - |
| TTA-C | 30.21/0.835 | 25.50/0.673 | 27.79/0.772 | 26.37/0.559 | 23.57/0.473 | 16.40/0.298 | 30.25/0.783 | 30.91/0.849 | 26.38/0.655 | 13.59 |
| SRTTA (ours) | 31.07/0.869 | 25.86/0.674 | 29.01/0.815 | **29.65/0.762** | 26.69/0.637 | 16.15/0.284 | **32.33/0.873** | 31.30/0.857 | 27.76/0.721 | 5.38 |
| SRTTA-lifelong (ours) | 31.07/0.869 | 25.83/0.674 | 29.18/0.819 | 29.48/0.797 | **27.10/0.673** | 16.27/0.273 | 31.71/0.864 | 31.22/0.853 | 27.73/**0.728** | 5.38 |

Table B: We report the PSNR/SSIM results of all corruption fields in DIV2K-C for $4\times$ SR.

| Methods | GaussianBlur | DefocusBlur | GlassBlur | GaussianNoise | PossionNoise | ImpulseNoise | SpeckleNoise | JPEG | Mean | GPU Time |
|---|---|---|---|---|---|---|---|---|---|---|
| | PSNR/SSIM | PSNR/SSIM | PSNR/SSIM | PSNR/SSIM | PSNR/SSIM | PSNR/SSIM | PSNR/SSIM | PSNR/SSIM | PSNR/SSIM | (seconds/image) |
| Bicubic | 25.83/0.718 | 24.10/0.641 | 25.35/0.699 | 23.17/0.500 | 22.15/0.475 | 15.1/0.384 | 25.29/0.658 | 25.07/0.681 | 23.27/0.595 | - |
| Real-ESRNet [22] | 26.82/0.765 | 25.17/0.704 | 26.75/0.762 | 25.49/**0.701** | 25.06/**0.692** | **19.24/0.509** | 26.47/0.749 | 25.70/0.720 | 25.09/**0.700** | 1.12 |
| SwinIR [15] | 28.48/0.785 | 25.81/0.692 | 27.44/0.753 | 22.96/0.454 | 21.40/0.420 | 14.59/0.225 | 26.66/0.670 | 27.25/0.731 | 24.32/0.591 | 8.15 |
| IPT [5] | 26.98/0.760 | 24.36/0.660 | 25.98/0.726 | 20.94/0.385 | 19.30/0.357 | 12.86/0.270 | 24.89/0.628 | 25.22/0.685 | 22.57/0.559 | 35.29 |
| HAT [6] | 27.09/0.764 | 24.37/0.660 | 26.01/0.728 | 20.38/0.368 | 18.76/0.344 | 11.65/0.177 | 24.70/0.623 | 25.11/0.680 | 22.26/0.543 | 5.95 |
| DAN [12] | 28.71/0.809 | 25.02/0.679 | 28.88/0.812 | 21.79/0.414 | 20.26/0.387 | 8.70/0.110 | 25.10/0.631 | 22.96/0.566 | 22.68/0.551 | 1.18 |
| DCLS-SR [18] | **30.38/0.834** | **26.48/0.709** | **30.58/0.838** | 24.43/0.525 | 22.90/0.479 | 6.94/0.038 | 27.27/0.696 | 27.46/0.736 | 24.56/0.607 | 1.47 |
| ZSSR [20] | 27.84/0.763 | 25.83/0.691 | 27.34/0.745 | 24.26/0.543 | 23.04/0.500 | 17.75/0.402 | 26.72/0.700 | 27.03/0.727 | 24.97/0.634 | 117.34 |
| KernelGAN [1]+ZSSR | 26.04/0.754 | 25.84/0.696 | 26.75/0.755 | 20.64/0.427 | 19.63/0.407 | 16.58/0.361 | 22.50/0.578 | 23.36/0.663 | 22.67/0.580 | 417.80 |
| MZSR [7] | 25.76/0.722 | 25.05/0.676 | 25.77/0.712 | 22.38/0.471 | 21.37/0.429 | 16.46/0.342 | 24.20/0.621 | 25.09/0.695 | 23.26/0.584 | 2.14 |
| SRDiff [14] | 26.52/0.746 | 24.18/0.649 | 25.92/0.723 | 16.25/0.180 | 15.50/0.172 | 12.23/0.172 | 19.41/0.356 | 24.18/0.649 | 20.52/0.456 | 72.22 |
| EDSR [16] | 28.31/0.780 | 25.81/0.692 | 27.40/0.751 | 23.49/0.479 | 22.10/0.443 | 15.28/0.283 | 26.80/0.676 | 27.34/0.734 | 24.57/0.605 | - |
| TTA-C | 28.19/0.776 | 25.76/0.691 | 27.29/0.747 | 24.03/0.504 | 22.71/0.468 | 16.38/0.357 | 27.03/0.685 | 27.45/0.736 | 24.85/0.621 | 20.11 |
| SRTTA (ours) | 28.61/0.792 | 26.24/0.702 | 28.09/0.775 | **26.58/0.684** | 25.27/0.617 | 15.73/0.318 | **28.24/0.763** | 27.66/0.742 | **25.80/0.674** | 4.47 |
| SRTTA-lifelong (ours) | 28.61/0.792 | 26.25/0.701 | 28.18/0.776 | 26.43/0.699 | **25.56/0.658** | 15.92/0.312 | 27.74/0.757 | 27.61/0.740 | 25.79/**0.679** | 4.47 |

## E.2  More Results on Test domains with Mixed Multiple Degradations.

In this part, we evaluate our SRTTA on DIV2K-MC, which consists of four test domains with mixed multiple degradations. In Table C, our SRTTA achieves the best performance on 4 domains with different mixed degradations, e.g., 0.619 (DualSR) → 0.775 (our SRTTA-lifelong) regarding the average SSIM metric. These results further validate the effectiveness of our proposed methods.

Table C: Comparison results with prior methods on DIV2K-MC. We report the PSNR($\uparrow$)/SSIM($\uparrow$) values of different methods.

| Methods | BlurNoise | BlurJPEG | NoiseJPEG | BlurNoiseJPEG | Mean |
|---|---|---|---|---|---|
| SwinIR [15] | 20.91/0.311 | 26.83/0.748 | 23.86/0.523 | 22.77/0.450 | 23.59/0.508 |
| IPT [5] | 21.28/0.327 | 26.83/0.748 | 24.15/0.535 | 22.96/0.459 | 23.81/0.517 |
| HAT [6] | 23.41/0.399 | 28.86/0.788 | 25.69/0.572 | 24.42/0.502 | 25.59/0.565 |
| DAN [12] | 24.14/0.438 | 28.95/0.791 | 26.20/0.593 | 24.82/0.519 | 26.03/0.585 |
| DCLS-SR [18] | 23.84/0.420 | 28.93/0.790 | 26.37/0.599 | 24.92/0.523 | 26.02/0.583 |
| ZSSR [20] | 24.95/0.493 | 29.02/0.793 | 26.68/0.617 | 25.24/0.542 | 26.47/0.611 |
| KernelGAN [1]+ZSSR | 23.08/0.424 | 28.32/0.786 | 21.90/0.474 | 22.76/0.443 | 24.02/0.532 |
| MZSR [7] | 18.73/0.213 | 24.90/0.667 | 20.37/0.398 | 20.62/0.354 | 21.16/0.408 |
| DualSR [8] | 25.59/0.561 | 28.24/0.787 | 23.78/0.586 | 24.62/0.541 | 25.56/0.619 |
| DDNM [24] | 22.62/0.389 | 26.82/0.746 | 25.11/0.582 | 23.81/0.504 | 24.59/0.555 |
| EDSR [16] | 24.02/0.430 | 28.93/0.790 | 26.08/0.587 | 24.73/0.514 | 25.94/0.580 |
| TTA-C | 24.29/0.446 | 28.93/0.790 | 26.35/0.598 | 24.91/0.522 | 26.12/0.589 |
| SRTTA (ours) | 26.93/0.709 | 28.93/**0.798** | **29.13/0.784** | 27.12/0.728 | 28.02/0.755 |
| SRTTA-lifelong (ours) | **27.67/0.749** | **29.02**/0.793 | 29.70/0.810 | **27.52/0.747** | **28.48/0.775** |

## E.3  More Results of Ablation Studies

**Effect of each component.** In this part, we investigate the effect of each component and provide more ablation studies. As shown in Table D, the baseline without the degradation classifier, generates the second-order degraded images with random degradation types, achieving a limited performance in terms of both PSNR and SSIM. The baseline without the adaptation consistency loss $\mathcal{L}_a$ results in the model collapse due to the lack of the consistency constraint. Without the self-supervised adaptation loss $\mathcal{L}_s$, the TTA performance of the adapted model drops significantly. These experimental results demonstrate the effectiveness of each component of our framework.

Table D: We report the PSNR/SSIM results of ablation studies of different components for $2\times$ SR.

| Methods | GaussianBlur | DefocusBlur | GlassBlur | GaussianNoise | PossionNoise | ImpulseNoise | SpeckleNoise | JPEG | Mean |
|---|---|---|---|---|---|---|---|---|---|
| | PSNR/SSIM | PSNR/SSIM | PSNR/SSIM | PSNR/SSIM | PSNR/SSIM | PSNR/SSIM | PSNR/SSIM | PSNR/SSIM | PSNR/SSIM |
| SRTTA (ours) | **31.07/0.869** | 25.86/0.674 | **29.01/0.815** | **29.66/0.762** | 26.69/0.637 | 16.15/0.284 | 32.33/0.873 | 31.30/0.857 | **27.76/0.721** |
| - w/o classifier $C(\cdot)$ | 29.43/0.812 | 25.51/0.675 | 27.51/0.756 | 22.05/0.546 | 25.64/0.571 | 15.66/0.260 | 31.50/0.836 | 31.19/0.855 | 26.06/0.664 |
| - w/o $\mathcal{L}_s$ | 30.65/0.854 | **25.87/0.680** | 28.43/0.795 | 28.04/0.644 | 24.83/0.534 | 15.96/0.274 | 31.96/0.847 | 31.47/0.862 | 27.15/0.686 |
| - w/o $\mathcal{L}_a$ | 12.29/0.254 | 5.67/0.397 | 5.65/0.403 | 12.87/0.477 | 10.29/0.072 | 11.48/0.213 | 11.67/0.218 | 11.99/0.477 | 10.24/0.314 |

**Effect of the hyperparameter $\alpha$ in Eqn. (6).** In this part, we investigate the effect of the weight of adaptation consistency loss $\alpha$. As shown in Table E, the adapted model with $\alpha = 1$ achieves the best TTA performance. Thus, we set the $\alpha = 1$ by default for our SRTTA during adaptation.

Table E: We report the PSNR/SSIM results of ablation studies of $\alpha$ for $2\times$ SR.

| $\alpha$ | GaussianBlur | DefocusBlur | GlassBlur | GaussianNoise | PossionNoise | ImpulseNoise | SpeckleNoise | JPEG | Mean | Set5 |
|---|---|---|---|---|---|---|---|---|---|---|
| | PSNR/SSIM | PSNR/SSIM | PSNR/SSIM | PSNR/SSIM | PSNR/SSIM | PSNR/SSIM | PSNR/SSIM | PSNR/SSIM | PSNR/SSIM | PSNR/SSIM |
| 0 | 12.29/0.254 | 5.67/0.397 | 5.65/0.403 | 12.87/0.477 | 10.29/0.072 | 11.48/0.213 | 11.67/0.218 | 11.99/0.477 | 10.24/0.314 | 11.42/0.421 |
| 0.1 | 11.11/0.318 | 8.02/0.043 | 5.65/0.403 | 22.78/0.683 | 5.65/0.403 | 12.50/0.484 | 15.03/0.542 | 30.95/0.850 | 13.96/0.466 | 37.66/0.959 |
| 0.5 | 10.47/0.134 | 10.46/0.313 | 23.69/0.708 | 29.21/**0.797** | 27.80/**0.717** | **16.30/0.290** | 32.10/0.871 | 30.98/0.850 | 22.63/0.585 | **37.75/0.959** |
| 1 | **31.07/0.869** | 25.86/0.674 | **29.01/0.815** | **29.66/0.762** | 26.69/0.637 | 16.15/0.284 | **32.33/0.873** | 31.30/0.857 | **27.76/0.721** | 34.59/0.924 |
| 2 | 30.86/0.862 | 25.91/0.678 | 28.71/0.804 | 29.04/0.710 | 25.89/0.591 | 16.06/0.279 | 32.23/0.865 | 31.45/0.861 | 27.52/0.706 | 35.41/0.933 |
| 5 | 30.74/0.857 | **25.91/0.680** | 28.53/0.798 | 28.47/0.670 | 25.29/0.558 | 16.00/0.276 | 32.08/0.855 | 31.48/0.862 | 27.31/0.695 | 35.89/0.939 |

**Effect of the hyperparameter $\rho$ in Eqn. (9).** In this part, we analyze the effect of the hyperparameter $\rho$, which decides the ratio of parameters to freeze, for the test-time adaptation. In Table F, when $\rho = 0.50$, our SRTTA achieves the best TTA performance on the DIV2K-C dataset on average in the lifelong setting. Meanwhile, we also investigate the effect of the adaptive parameter preservation (APP) strategy in the parameter-reset setting. As shown in Table F, our APP strategy (with $\rho = 0.50$)

merely has little impact on the TTA in the parameter-reset setting. These results demonstrate the effectiveness of the APP strategy in test-time adaptation for image super-resolution.

Table F: We report the PSNR/SSIM results of ablation studies of $\rho$ for $2\times$ SR in the parameter-reset and lifelong setting, our model is SRTTA and SRTTA-lifelong.

| Setting | 0 | 0.1 | 0.2 | 0.3 | 0.5 | 0.7 | 0.9 | 1 |
|---|---|---|---|---|---|---|---|---|
| SRTTA | 27.74/0.729 | 27.79/0.730 | **27.82/0.729** | 27.82/0.727 | 27.76/0.721 | 27.55/0.709 | 27.08/0.682 | 26.21/0.645 |
| SRTTA-lifelong | 27.46/0.726 | 27.60/0.727 | 27.66/0.728 | 27.72/0.728 | **27.73/0.728** | 27.73/0.725 | 27.50/0.706 | 26.21/0.645 |

**Comparison with other anti-forgetting methods.** In this part, we compare our adaptive parameter preservation (APP) strategy with two baseline methods to demonstrate the effectiveness of our strategy in preserving the learned knowledge of pre-trained SR models. **Stochastic Restoration (STO)** [21] randomly selects a different set of parameters (with a ratio of $1\%$) and restores them back to the parameters of the pre-trained models. **Random Selection (RS)** selects a fixed set of parameters before adaption and freezes them not to update. As shown in Table G, our APP strategy achieves the best TTA results on the DIV2K-C dataset. Meanwhile, with the same ratio of selected parameters, our APP strategy consistently outperforms the Random Selection baseline for the anti-forgetting. These results demonstrate that our adaptive selection is able to select the important parameters and preserve the knowledge of the pre-trained model.

Table G: We report the PSNR/SSIM results of ablation studies of adaptive parameter preservation (APP) strategy for $2\times$ SR in the lifelong setting.

| Dataset | STO [21] | RS with different $\rho$ | | | APP with different $\rho$ (ours) | | |
|---|---|---|---|---|---|---|---|
| | | 0.3 | 0.5 | 0.7 | 0.3 | 0.5 | 0.7 |
| DIV2K-C (with degradation shift) | 27.17/0.687 | 27.52/0.727 | 27.62/0.728 | 27.68/0.726 | 27.72/0.728 | **27.73/0.728** | 27.73/0.725 |
| Set5 (w/o degradation shift) | **35.57/0.938** | 33.95/0.913 | 34.02/0.914 | 34.24/0.918 | 34.11/0.916 | 34.23/0.917 | 34.38/0.920 |

## E.4 Effectiveness on a New Unknown Domain

In this part, we further evaluate our SRTTA on a new unknown domain with the degradation of processed camera sensor noise [3, 26], which is not used in the training phase of the SR model or that of the degradation classifier. We report the PSNR($\uparrow$) and SSIM($\uparrow$) values of different methods on 100 images with random processed camera sensor noise. In Table H, our SRTTA method is also able to improve the model performance on this unknown degradation. These experimental results further demonstrate the generalization capability of our SRTTA model to unknown degradation types.

## E.5 Comparison with Patch-Recurrence Reconstruction Loss

In this part, we investigate the effect of our second-order reconstruction loss. We compare our loss with the loss of existing zero-shot methods [20, 7, 1]. Based on the assumption of patch recurrence across scales [9, 29], these methods downsample the test image to obtain an image with a lower resolution and reconstruct the test image from the downsampled image. For simplicity, we call this patch-recurrence loss. For a fair comparison, we further downsample the second-order degraded images that are obtained using our second-order degradation scheme and reconstruct the test image with the patch-recurrence loss. As shown in Table I, our SRTTA with our second-order reconstruction loss consistently outperforms the baseline with the patch-recurrence loss. These results demonstrate the effectiveness of our second-order reconstruction loss.

Table H: Results of different methods on the unknown domain with the degradation of processed camera sensor noise.

| Methods | PSNR | SSIM |
|---|---|---|
| SwinIR [15] | 19.45 | 0.496 |
| IPT [5] | 19.51 | 0.500 |
| HAT [6] | 21.52 | 0.596 |
| DDNM [24] | 19.63 | 0.518 |
| DAN [12] | 21.53 | 0.598 |
| DCLS-SR [18] | 21.57 | 0.605 |
| DualSR [8] | 21.14 | 0.586 |
| MZSR [7] | 20.40 | 0.438 |
| ZSSR [20] | 21.57 | 0.621 |
| KernelGAN [1]+ZSSR | 20.60 | 0.543 |
| EDSR [16] | 21.56 | 0.601 |
| SRTTA (ours) | **21.81** | **0.647** |

## E.6 Effect of the Feature-level Reconstruction

In this part, we investigate the effect of different reconstruction levels. In our second-order reconstruction, we use the feature-level reconstruction to adapt the pre-trained model as in Eqn. (6). We

Table I: We report the PSNR/SSIM results of ablation studies of the patch-recurrence reconstruction loss for 2×SR in parameter-reset and lifelong settings.

| methods | Setting | GaussianBlur PSNR/SSIM | DefocusBlur PSNR/SSIM | GlassBlur PSNR/SSIM | GaussianNoise PSNR/SSIM | PossionNoise PSNR/SSIM | ImpulseNoise PSNR/SSIM | SpeckleNoise PSNR/SSIM | JPEG PSNR/SSIM | Mean PSNR/SSIM |
|---|---|---|---|---|---|---|---|---|---|---|
| Patch-recurrence loss | parameter-reset | 29.87/0.828 | 25.51/0.674 | 27.67/0.767 | 28.55/0.680 | 26.41/0.605 | 19.72/0.407 | 31.71/0.842 | 31.02/0.853 | 27.56/0.707 |
| Patch-recurrence loss | lifelong | 29.87/0.828 | 25.51/0.674 | 27.67/0.767 | 28.52/0.678 | 26.40/0.604 | **20.48/0.430** | 31.76/0.846 | 31.01/0.853 | 27.65/0.710 |
| SRTTA(ours) | parameter-reset | 31.07/0.869 | **25.86**/0.674 | 29.01/0.815 | **29.66**/0.762 | 26.69/0.637 | 16.15/0.284 | **32.33/0.873** | **31.30/0.857** | **27.76**/0.721 |
| SRTTA(ours) | lifelong | **31.07/0.869** | 25.83/0.674 | **29.18/0.819** | 29.48/0.797 | **27.10/0.673** | 16.27/0.273 | 31.71/0.864 | 31.22/0.853 | 27.73/**0.728** |

Table J: We report the PSNR/SSIM results of ablation studies of feature-level and image-level reconstruction for 2× SR in the lifelong setting.

| Reconstruct-level | Scale | GaussianBlur PSNR/SSIM | DefocusBlur PSNR/SSIM | GlassBlur PSNR/SSIM | GaussianNoise PSNR/SSIM | PossionNoise PSNR/SSIM | ImpulseNoise PSNR/SSIM | SpeckleNoise PSNR/SSIM | JPEG PSNR/SSIM | Mean PSNR/SSIM |
|---|---|---|---|---|---|---|---|---|---|---|
| EDSR [16] | 2 | 30.28/0.837 | 25.52/0.673 | 27.82/0.773 | 25.87/0.536 | 22.96/0.449 | 15.87/0.269 | 30.52/0.778 | 30.83/0.847 | 26.21/0.645 |
| Image-level | 2 | **31.21/0.871** | 25.73/0.671 | 29.16/0.817 | **29.56**/0.796 | 26.74/0.654 | 16.24/0.263 | **32.47/0.884** | **31.44/0.860** | **27.82**/0.727 |
| Feature-level | 2 | 31.07/0.869 | **25.83/0.674** | **29.18/0.819** | 29.48/**0.797** | **27.10/0.673** | **16.27/0.273** | 31.71/0.864 | 31.22/0.853 | 27.73/**0.728** |
| EDSR [16] | 4 | 28.31/0.780 | 25.81/0.692 | 27.40/0.751 | 23.49/0.479 | 22.10/0.443 | 15.28/0.283 | 26.80/0.676 | 27.34/0.734 | 24.57/0.605 |
| Image-level | 4 | 28.61/0.790 | 26.23/0.697 | 28.13/0.773 | 26.46/0.685 | 25.28/0.623 | **15.72/0.297** | **28.02/0.751** | **27.74/0.746** | **25.77**/0.670 |
| Feature-level | 4 | **28.78/0.795** | **26.31/0.703** | **28.16/0.776** | **26.28/0.691** | 25.46/0.643 | 15.62/0.294 | 27.77/**0.755** | 27.61/0.740 | 25.75/**0.675** |

compare a baseline with an image-level reconstruction, which means we reconstruct the output of the SR model instead of the feature in the middle layer. As shown in Table J, when reconstructing at the image level, the adapted model achieves a comparable performance for both 2× SR and 4× SR. Thus, both reconstruction levels are optional, we use the feature-level reconstruction by default.

Table K: We report the PSNR/SSIM results of ablation studies of adapted iterations for 2× SR in the parameter-reset setting.

| Iterations | GaussianBlur PSNR/SSIM | DefocusBlur PSNR/SSIM | GlassBlur PSNR/SSIM | GaussianNoise PSNR/SSIM | PossionNoise PSNR/SSIM | ImpulseNoise PSNR/SSIM | SpeckleNoise PSNR/SSIM | JPEG PSNR/SSIM | Mean PSNR/SSIM | GPU Time seconds/image |
|---|---|---|---|---|---|---|---|---|---|---|
| 1 | 30.49/0.844 | 25.59/0.675 | 28.08/0.782 | 27.05/0.592 | 24.34/0.507 | 15.98/0.279 | 31.32/0.810 | 31.09/0.853 | 26.74/0.668 | 0.60 |
| 2 | 30.72/0.854 | 25.82/**0.679** | 28.62/0.800 | 27.76/0.630 | 24.99/0.538 | 16.02/0.280 | 31.76/0.830 | 31.23/0.856 | 27.12/0.683 | 1.41 |
| 5 | 30.64/0.861 | 25.80/0.677 | 28.88/0.811 | 28.90/0.702 | 26.10/0.600 | 15.98/0.278 | 32.19/0.861 | **31.37/0.859** | 27.48/0.706 | 2.73 |
| 10 | 31.07/**0.869** | **25.86**/0.674 | 29.01/0.815 | **29.66/0.762** | **26.69/0.637** | **16.15/0.284** | **32.33/0.873** | 31.30/0.857 | **27.76/0.721** | 5.38 |
| 20 | **31.15/0.870** | 25.80/0.675 | **29.32/0.824** | 29.24/**0.772** | 26.05/0.601 | 16.08/0.276 | 32.14/0.865 | 31.31/0.856 | 27.64/0.717 | 10.82 |
| 50 | 30.71/0.856 | 25.64/0.674 | 29.28/0.821 | 29.12/0.743 | 25.40/0.569 | 16.02/0.265 | 31.54/0.838 | 28.46/0.846 | 27.02/0.701 | 25.64 |

## E.7 Effect of Adaptation Iterations for Each Image

In this part, we investigate the effect of different adaptation iterations for each image. As shown in Table K, we compare the TTA performance of SRTTA with a different number of iterations $S$ for each image. When the number of iterations $S$ is small, the adapted SR model is not able to learn how to remove the degradation for these images well. When the number of iterations $S$ is too large, the performance improvement of SRTTA diminishes and the adaptation cost will be greatly increased. Thus, we set $S$ to 10 for a better efficiency-accuracy trade-off.

## E.8 The Statistics of the Degradation Types of Real-world Images

In this part, we count the statistics of the degradation types of real-world images and report the results in Table L. We use our degradation classifier to identify the degradation types of real-world images from five datasets, including RealSR [4], DRealSR [25], DPED [13], OST300 [23] and ADE20K [28]. Experimental results in Table L show that the degradation type of blur happens the most among these real-world datasets.

## E.9 Further Analysis of the Domain Shift Issue

In this part, we investigate the effect of the domain shift issues for pre-trained SR models, which are trained on specific domains with different degradation types. We use the EDSR baseline model as the model for analysis. In total, we separately train four EDSR baseline models on clean, blur, noise and JPEG domains, respectively. The corresponding four models are named EDSR, EDSR-B, EDSR-N and EDSR-J, respectively. We evaluate these four models on clean images and the test images with Gaussian Blur, Gaussian Noise or JPEG degradations in Figure B and Figure C.

As shown in Figure B and Figure C, when domain shift occurs, the pre-trained EDSR models, which are trained on domains different from the test domains, cannot remove the degradation from test

Table L: The count of the predicted degradation types of the real-world images from the five datasets. Note that some images can contain more than one degradation type simultaneously.

| Dataset (# Images) | Clean | Blur | Noise | JPEG |
|---|---|---|---|---|
| RealSR (912) | 48 | 860 | 1 | 149 |
| DRealSR (35148) | 1378 | 33770 | 0 | 1 |
| DPED (187) | 103 | 33 | 21 | 64 |
| OST300 (300) | 52 | 23 | 14 | 221 |
| ADE20K (27574) | 0 | 3452 | 2424 | 27573 |
| Total (64121) | 1581 | **38138** | 2460 | 28008 |

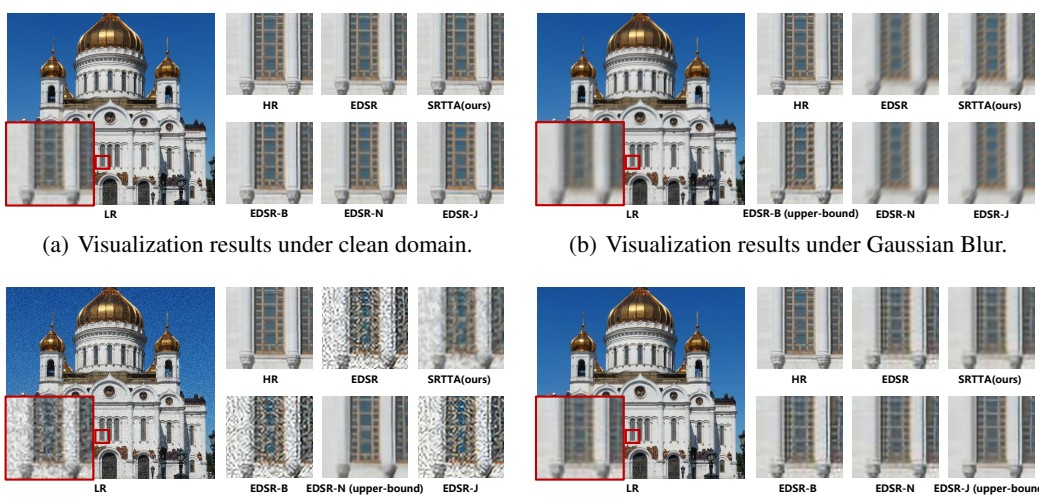

(a) Visualization results under clean domain.

(b) Visualization results under Gaussian Blur.

(c) Visualization results under Gaussian Noise.

(d) Visualization results under JPEG.

Figure B: Visualization of the domain shift issue under different domains for $2\times$ SR.

images and generate unsatisfactory HR images with artifacts. For example, EDSR-B models cannot remove the noise and JPEG degradation, the EDSR-N and EDSR-J are also unable to remove the blur degradation. Instead, after test-time adaptation, our SRTTA is capable of handling the test images with unknown degradations and generating HR images with fewer artifacts. For example, our SRTTA is able to generate sharper HR images than EDSR-N and EDSR-J under Gaussian Blur domains. Indeed, our SRTTA models may be unable to completely remove the degradation compared with the upper-bound models, such as EDSR-B under Gaussian Blur. Thus, these drawbacks are required to be further addressed in future works.

# F   Visualization Results

## F.1   Visualization Results on the DIV2K-C Images

In this part, we show more visualization comparison results of different SR methods on test images of the DIV2K-C dataset for both $2\times$ and $4\times$ SR. As shown in Figures D and E, our SRTTA is able to reduce the degradation from the test images and generate more plausible HR images.

## F.2   Visualization Results on the Real-World Images

In this part, we conduct a comprehensive comparison of our SRTTA with existing approaches on two real-world datasets, including DPED [13], ADE20K [28] and OST300 [23]. As shown in Figures F and G, our SRTTA methods consistently generate more satisfactory HR images with less degradation of unknown noise or artifacts.

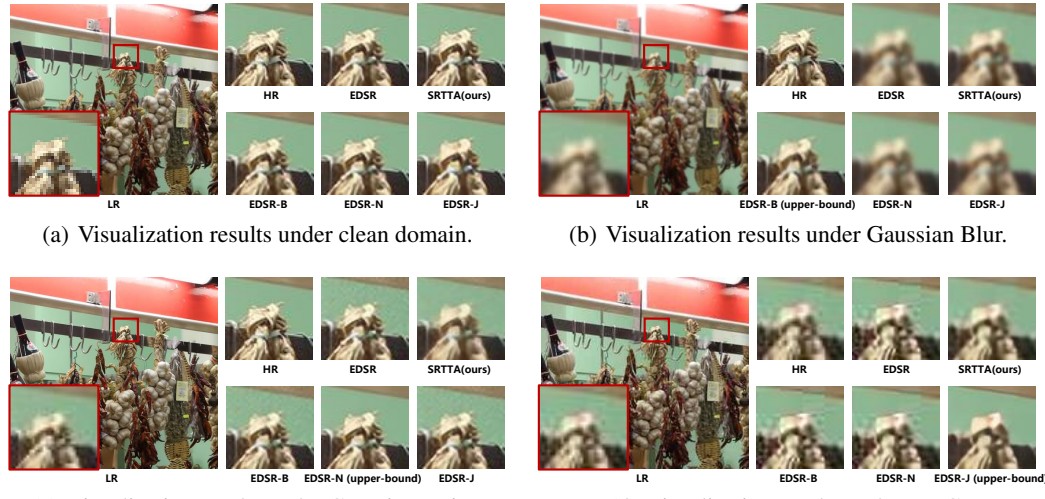

(a) Visualization results under clean domain.

(b) Visualization results under Gaussian Blur.

(c) Visualization results under Gaussian Noise.

(d) Visualization results under JPEG.

Figure C: Visualization of the domain shift issue under different domains for 2× SR.

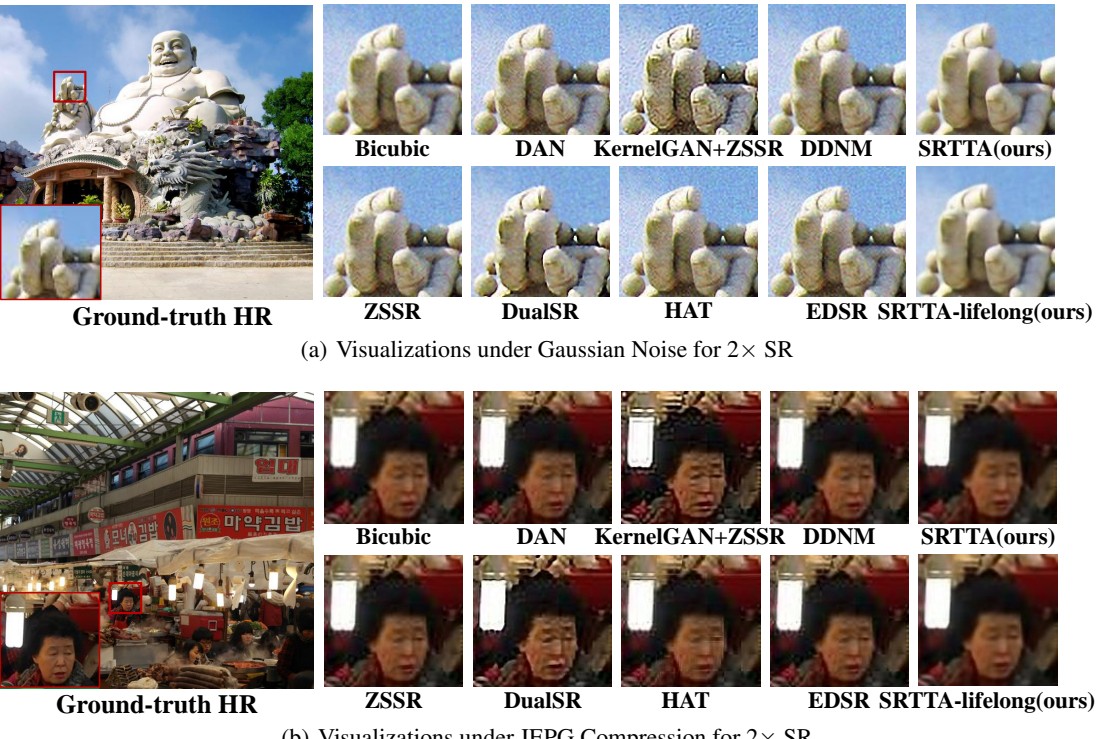

(a) Visualizations under Gaussian Noise for 2× SR

(b) Visualizations under JEPG Compression for 2× SR

Figure D: Visualization comparison on DIV2K-C test images with degradation for 2× SR.

## G  Limitation Analysis and Border Impacts

### G.1  Limitation Analysis

In this part, we analyze the limitations of our SRTTA and existing SR methods. When test images are corrupted at a high level, our SRTTA may not be able to completely remove the degradation and result in generated HR images with the existing degradation. For example, we show more visualization results of different methods on test images with Impulse Noise degradation in Figure H.

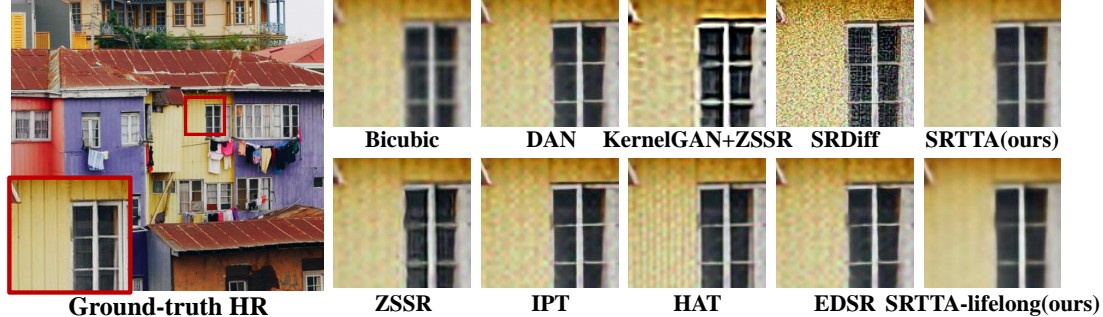

(a) Visualizations under PossionNoise for 4× SR

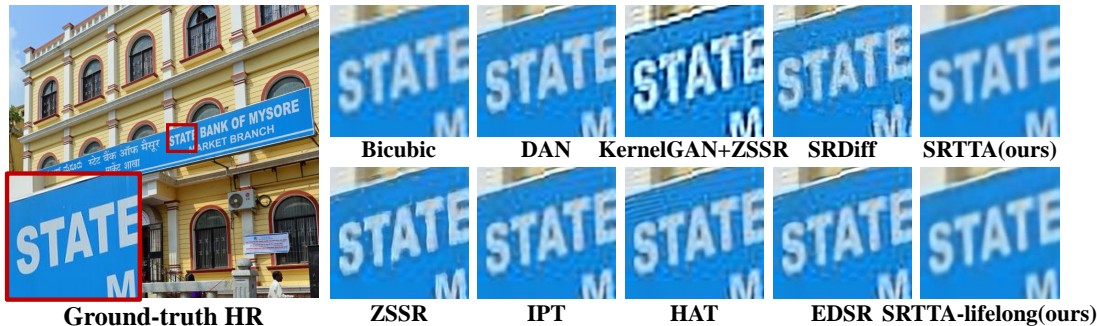

(b) Visualizations under JEPG Compression for 4× SR

Figure E: Visualization comparison on DIV2K-C test images with degradation for 4× SR.

Since Real-ESRGAN [22] uses several different degradation types to construct training data, this model is able to remove the degradation in many cases. However, this model still suffers from the degradation shift issue, such as it cannot remove the gray Impulse noise from the test images as shown in Figure H(a). Moreover, Real-ESRGAN may generate HR images with over-smooth regions when removing the noise degradation and introduce some unpleasant artifacts due to the GAN training [22], which are shown in Figure H(b) and Figure H(c), respectively. Although our SRTTA cannot also completely remove the degradation from the test images in these cases, our SRTTA often preserves the original information of the test images. These results show the limitations of our SRTTA and existing methods and have a great impact on the practical application. Thus, these drawbacks are in urgent need to address in future works.

## G.2 Broader Impacts

Our proposed SRTTA method is capable of improving the resolution of low-resolution test images in real-world applications, resulting in enhanced image clarity and enabling a precise understanding of image content. However, it is important to exercise caution during aggressive TTA adaptation, as this may result in the introduction of artifacts or distortions that have the potential to negatively impact downstream analyses such as microscopy, remote sensing, and surveillance.

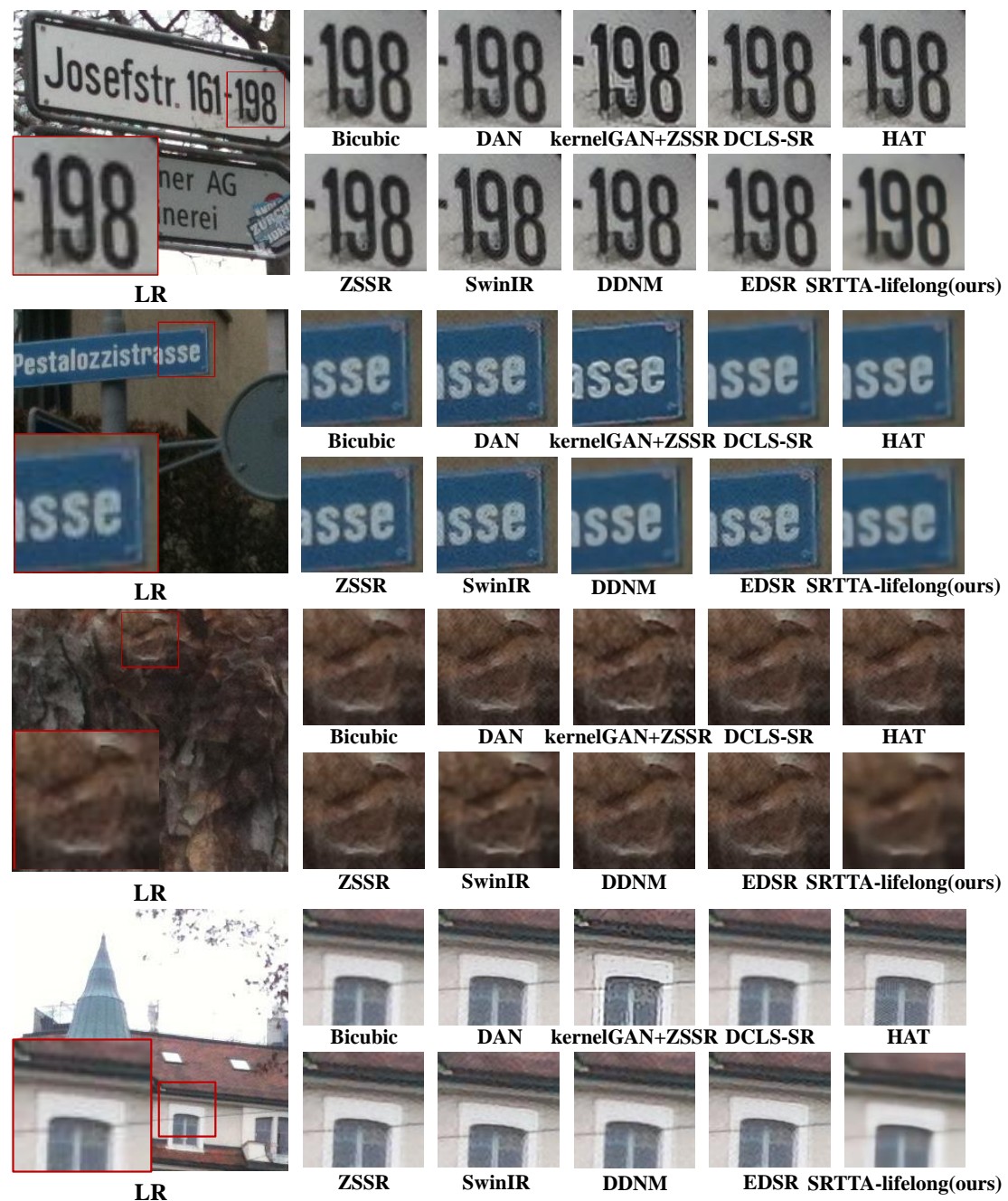

Figure F: Visualization comparison of different methods on real-world test images from DPED [13].

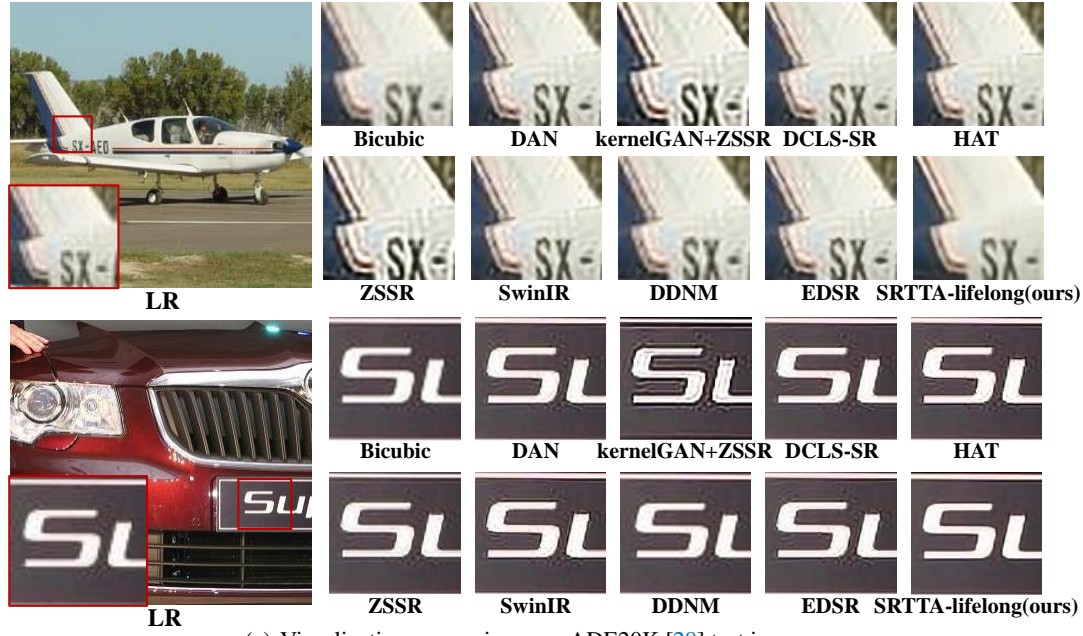

(a) Visualization comparisons on ADE20K [28] test images.

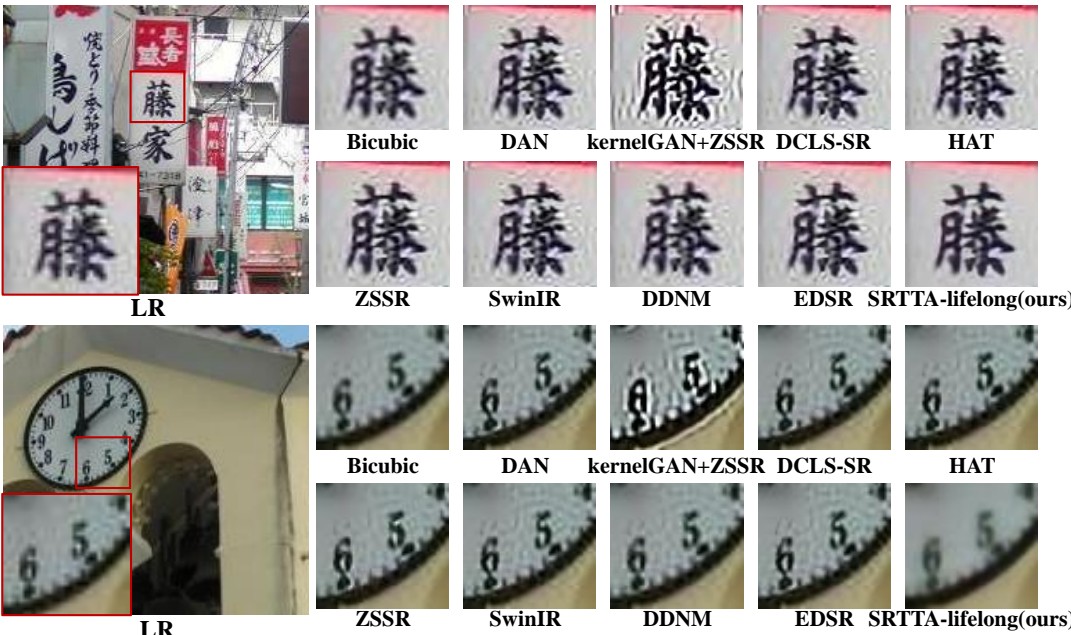

(b) Visualization comparisons on OST300 [23] test images.

Figure G: Visualization comparison on real-world test images from for $2\times$ SR.

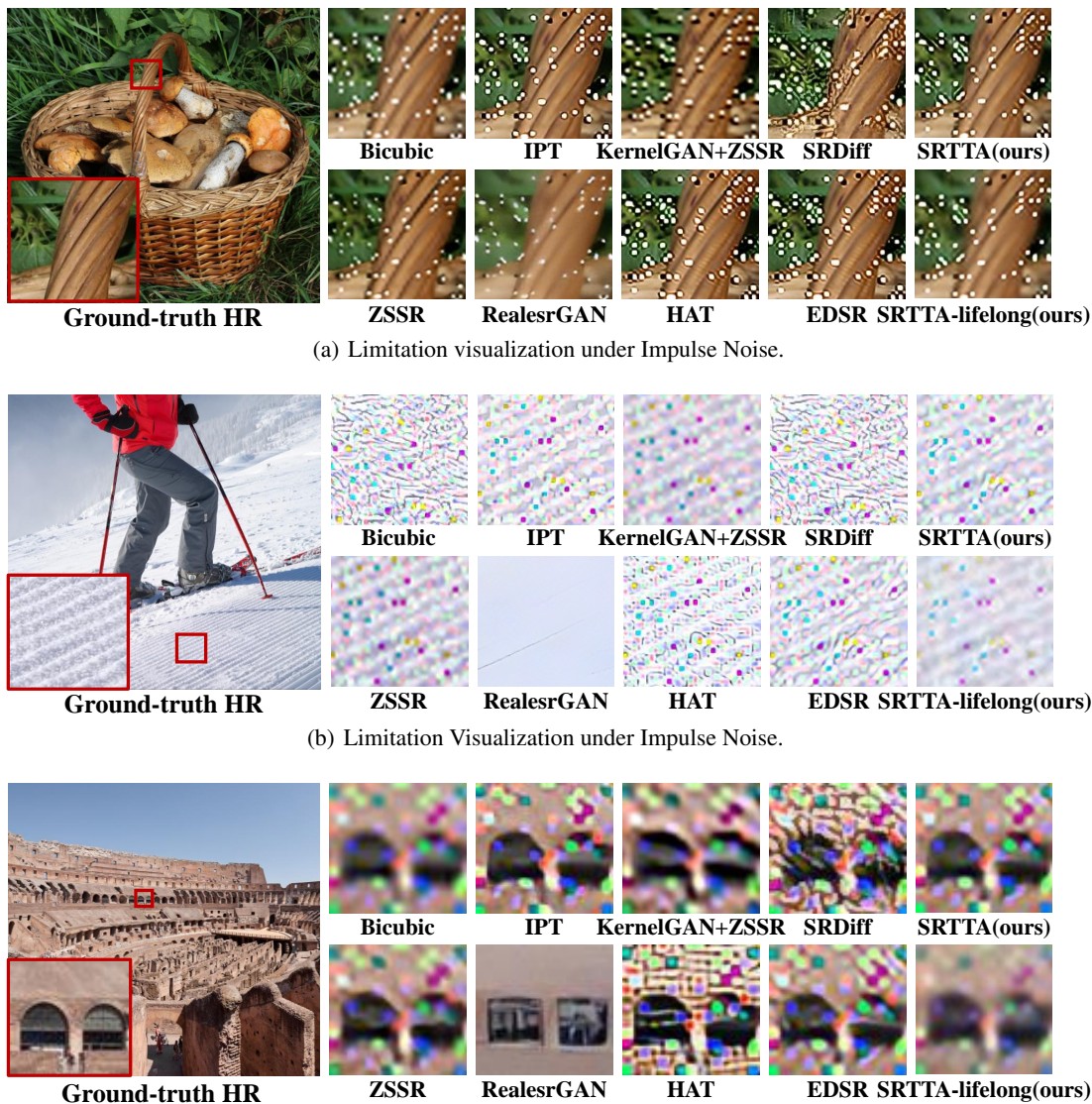

(a) Limitation visualization under Impulse Noise.

(b) Limitation Visualization under Impulse Noise.

(c) Limitation visualization under Impulse Noise.

Figure H: Limitations visualization on DIV2K-C test images with degradation for $4\times$ SR.

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
