# OpenReview forum: "Efficient Test-Time Adaptation for Super-Resolution with Second-Order Degradation and Reconstruction"
_NeurIPS.cc/2023/Conference — NeurIPS 2023 poster_

### Official Review · Reviewer_QTZE · 2023-06-26

**Soundness:** 3 good
**Presentation:** 3 good
**Contribution:** 3 good
**Rating:** 7
**Confidence:** 4

**Summary:**

This paper proposes a fast test-time adaptation SR framework named SRTTA, which is capable of super-resolving images with various degradation types while maintaining efficiency. Specifically, the authors first exploit a pre-trained degradation classifier to predict the degradation type of the test image and construct the paired dataset based on the degradation. Then, the authors adapt the SR model by conducting the feature-level reconstruction learning from the test image and its second-order degraded image generated by the designed second-order degradation scheme. Finally, the authors design the second-order reconstruction loss to adapt the pre-trained model to test images with different degradation types.

After reviewing the rebuttal, I've noticed that most of my previous concerns have been addressed. Therefore, I would like to increase my initial score.

**Strengths:**

This paper provides an interesting image super-resolution idea for different degradations in the real-world test-time scenario. Compared with the classical deep network, it is applicable to any pretrained SR model. The proposed SRTTA can adapt a given SR model to estimate the degradation shifts in the test images. In addition, the authors construct a dataset named DIV2K-C with 8 different degradation types.

**Weaknesses:**

1. The proposed framework seems to be oriented towards a single degradation from different degradation types in the real-world test-time scenario. However, for the real-world scenario, the practical degradation is much more complex and could be the combinations of different degradations. This paper may not be applicable to this situation.
2. The proposed dataset also may not be suitable for the aforementioned situation.
3. The methods for visualization comparison are older compared to the methods for quantitative comparison.
4. The GPU Time of the proposed method in Table 1 is not the optimal.
5. The notation in section Super-Resolution Test-Time Adaptation should be consistent and continuous, such as x_t in line 149.

**Questions:**

1. The authors should clarify the 1st and 2nd concerns in section weaknesses. This seems to be an important reason why the proposed method is superior to other SOTA methods.
2.  A detailed explanation or citation for the domain shifts should be given in line 136 on page 3.
3. The authors should clarify the 3rd concern in section weaknesses and provide more convincing visualization.
4. The authors mention that some previous methods are quite time-consuming in the motivation, so the authors should clarify the 4th concern in section weaknesses.
5. How does the proposed network work when the test images are degraded by multiple degradation types at the same time?
6. The motivation mentioned from line 158 to line 160 in section 4.1 should be further demonstrated or cited.

**Limitations:**

The application scenario of the proposed method is limited for the single degradation.

---

> ### Author Rebuttal · Authors · 2023-08-09
>
> > Q1. Lack of evaluation on domains with mixed multiple degradations.
>
> A1: In our paper, we seek to adapt pre-trained SR models to target domains when real-world images contain different degradations, which can severely limit the application of most SR models. To this end, we propose a novel SRTTA framework to adapt SR models to domains with unseen degradations during testing.
>
> Our SRTTA framework is applicable to real-world test images with mixed multiple degradations by substituting the degradation classifier with a multi-label classifier. More details and experimental results can be referred to General Response 2. From the results, our SRTTA achieves the best performance on mixed multiple degradations, which demonstrates the effectiveness of our SRTTA framework in these complex scenarios.
>
> > Q2. The proposed dataset does not contain test images with mixed multiple degradations.
>
> A2. In the rebuttal, we further construct a dataset named **DIV2K-MC**, which consists of four domains with mixed multiple degradations, including **BlurNoise**, **BlurJPEG**, **NoiseJPEG**, and **BlurNoiseJPEG**. The test images in the **BlurNoiseJPEG** domain contain the combined degradation of blur, noise and JPEG simultaneously. Experimental results on this DIV2K-MC dataset further demonstrate the effectiveness of our SRTTA on domains with mixed multiple degradations. Please refer to General Response 2 for more experimental results.
>
> > Q3. The methods for visualization comparison are older compared to the methods for quantitative comparison.
>
> A3: We indeed have included comprehensive visualization comparisons of different methods in Figures 3, 4, and 5 of the supplementary materials. As shown in these figures, our SRTTA is able to reduce the degradation from the test images and generate more plausible HR images.
>
> > Q4. The GPU Time of the proposed method in Table 1 is not the optimal.
>
> A4: In our paper, we seek to adapt a pre-trained SR model to domains with unseen degradations during testing. Due to the additional time for test-time adaptation, our SRTTA cannot achieve the best inference speed. However, those methods that require fewer inference times are trained on domains with Gaussian blur degradation only, such as DAN, DCLS-SR, and MZSR. Consequently, they often suffer from the domain shift issue when applied to other domains, such as domains with Gaussian noise or JPEG degradations. Instead, with comparable efficiency, our SRTTA achieves an impressive improvement on average for all domains (Refer to Table 1 of the paper). In conclusion, our SRTTA achieves a better tradeoff between performance and efficiency.
>
> > Q5. Typos in line 149.
>
> A5: We will carefully check the notations in the revision.
>
> > Q6. Lack of relevant citations for the domain shifts.
>
> A6: We will discuss and cite the related reference [1,2,3] in the revised paper.
>
> [1] WILDS: A benchmark of in-the-wild distribution shifts, ICML 2021.
> [2] Continual test-time domain adaptation, CVPR 2022.
> [3] Blind image super-resolution: A survey and beyond, TPAMI 2022.
>
> > Q7. How does the proposed network work when the test images are degraded by multiple degradation types at the same time?
>
> A7: Our SRTTA is applicable to the domain with single degradation or mixed multiple degradations. To adapt SR models to the domains with mixed multiple degradations, we change the degradation classifier to a multiple-label classifier. The multi-label classifier predicts the probabilities of blur, noise and JPEG degradation simultaneously for each test image. With the predicted multiple degradation types, we degrade the test images $\mathbf{x}$ to generate corresponding second-order degraded images $\mathbf{x}_{sd}$ based on the classical degradation model [1]. Experimental results in Table A of General Response 2 demonstrate the effectiveness of our SRTTA on the domains with mixed multiple degradations.
>
> [1] Real-ESRGAN: Training real-world blind super-resolution with pure synthetic data, ICCV, 2021.
>
> > Q8. The motivation of the second-order degradation scheme should be further demonstrated or cited.
>
> A8: In real-world scenarios, when the domain shift issue occurs, the clean images corresponding to test images are often unavailable, making the SR models hard to learn how to remove the degradation from the test images. In this context, the key challenge is how to effectively construct (pseudo) paired data to adapt SR models to the target domain. To this end, the commonly used technique is to construct paired data with the second-order degradation scheme, which has been proven to be highly effective by previous works [1, 2].
>
> However, these methods [1, 2] primarily concentrate on estimating blur degradation while neglecting other degradations, such as noise and JPEG degradation. Moreover, these methods often require numerous iterations to accurately estimate the blur degradation model, leading to significant time consumption. In this work, we address these two limitations by proposing our SRTTA framework. We will provide a clearer explanation and the citation in the revised paper.
>
> [1] Blind super-resolution kernel estimation using an internal-GAN, NeurIPS 2019.
> [2] Flow-based kernel prior with application to blind super-resolution, CVPR 2021.

---

> > ### Comment · Reviewer_QTZE · 2023-08-15
> > **Good Rebuttal (with additional minor suggestions)**
> >
> > Thank you for your response. After reviewing the rebuttal, I've noticed that most of my previous concerns have been addressed. Therefore, I would like to increase my score. However, I still have a few more suggestions:
> > The details of the SRTTA-MC should be included in the revised paper.
> > Given that the inference speed of the proposed SRTTA isn't the fastest, it would be beneficial to include the analysis mentioned in A4 within the experiments. This could help clarify the reasons behind this aspect.
> > The hyperparameter $\rho$ should be explicitly incorporated into Equation (8). While the authors investigate the sensitivity of $\rho$ in Equation (8) in Table 4, the absence of a direct reference to $\rho$ within Equation (8) could potentially hinder a clear understanding of the relationship between the hyperparameter $\rho$ and the equation.

---

> > > ### Author Response · Authors · 2023-08-17
> > > **Responses to the additional minor suggestions.**
> > >
> > > Dear Reviewer QTZE,
> > >
> > > We would like to thank you for the new feedback and the increment in your score. We will carefully revise our paper based on the suggestions.
> > > We will (1) illustrate the details of the SRTTA-MC in Section 4.1 and (2) include the analysis of the inference speed in the experiments.
> > > (3) To better clarify the relationship between the hyperparameter $\rho$ and Equation (8), we will reformulate Equation (8) to explicitly incorporate $\rho$ into Equation (8) as follows:
> > >
> > > $ \mathcal{S} = \\{ \theta_i^0 \| \omega(\theta^0_i) > \tau_{\rho}, \theta_i^0 \in \theta^0 \\}, $
> > >
> > > where $\tau_{\rho}$ denotes the first $\rho$-ratio largest value obtained by ranking the value $\omega(\theta_i^0)$, $\rho$ is a hyperparameter to control the ratio of parameters to be frozen. $\theta^0$ is the parameters of the pre-trained SR model and $\theta^0_i$ is the $i$-th parameter within $\theta^0$.
> > >
> > > Sincerely,
> > > Authors of #2666

---

> ### Author Response · Authors · 2023-08-11
> **Looking forward to the response from Reviewer QTZE**
>
> Dear Reviewer QTZE,
> We would like to thank you for your valuable feedback to improve our work. We are wondering whether our response has addressed your questions and can improve your opinion of our work. Kindly let us know if you have any other concerns, and we will do our best to address them.
> Best regards,
> Authors of #2666

---

### Official Review · Reviewer_G7Pf · 2023-07-04

**Soundness:** 3 good
**Presentation:** 3 good
**Contribution:** 2 fair
**Rating:** 7
**Confidence:** 5

**Summary:**

The paper proposes a test-time adaptation framework called SRTTA for image super-resolution (SR) to address the degradation shift issue between training and test images. The SRTTA framework adapts a pre-trained SR model to different degradation types observed in real-world test images. It introduces a second-order degradation scheme that generates paired data based on the degradation type predicted by a pre-trained degradation classifier. The SR model is then adapted through feature-level reconstruction learning from the initial test image to its second-order degraded counterparts. Experimental results on a newly synthesized DIV2K-C dataset and a real-world DPED dataset demonstrate the effectiveness and practicality of the SRTTA framework.

**Strengths:**

The paper introduces a new framework, SRTTA, that allows rapid adaptation of a pre-trained SR model to various degradation types observed in test images. The second-order degradation scheme and feature-level reconstruction learning provide a practical and effective solution for adapting the model to real-world scenarios.

Unlike some existing methods that focus on a single degradation type, SRTTA considers multiple degradation types, such as blur, noise, and JPEG artifacts, making it more applicable in real-world scenarios where images can have diverse degradation types.

SRTTA achieves fast adaptation by using only a single degraded low-resolution image during testing. This makes it more practical for real-time applications compared to methods that require extensive iterations or GAN-based approaches for degradation estimation.

**Weaknesses:**

1. Although the paper claims to evaluate the framework on a real-world DPED dataset, the evaluation results and analysis are not sufficient, in the per. The lack of comprehensive evaluation on diverse real-world datasets limits the generalizability and applicability of the proposed framework.

2. The paper focuses on adapting the pre-trained SR model to different degradation types using second-order degradation. However, there is limited discussion or analysis on the generalization capability of the adapted model to unseen degradation types. It would be valuable to investigate the model's performance when faced with degradation types not encountered during training, or different types combine the degradations.

3. The paper primarily focuses on evaluating the performance of the proposed method in terms of PSNR (Peak Signal-to-Noise Ratio). While PSNR is a commonly used metric, it does not always correlate well with perceived image quality. The evaluation could have included other quality metrics, such as Mean Opinion Score (MOS) or other human eye-related metrics, to provide a more comprehensive assessment of the proposed method.

**Questions:**

refer to the weakness

**Limitations:**

yes, the authors have addressed some limitations of the paper

---

> ### Author Rebuttal · Authors · 2023-08-09
>
> > Q1. Evaluation merely on a real-world dataset limits the generalizability and applicability of the proposed framework.
>
> A1: Our SRTTA is able to remove the degradation of real-world images and generate images with fewer artifacts, as shown in Figure 3 of the paper and Figure 5 of the supplementary materials. To comprehensively demonstrate the effectiveness of our SRTTA, we further evaluate our SRTTA on two real-world datasets, including OST300[1] and ADE20K[2]. As shown in **Figure A of the supplementary PDF file accompanying this rebuttal**, results demonstrate the superiority of our SRTTA, resulting in the generated images with fewer artifacts.
>
> [1] Recovering realistic texture in image super-resolution by deep spatial feature transform, CVPR 2018.
> [2] Semantic understanding of scenes through the ADE20K dataset. IJCV 2019.
>
> >  Q2. There is limited discussion or analysis on the generalization capability of the adapted model to unseen degradation types.
>
> A2: Our SRTTA framework aims to adapt the pre-trained model to different domains with unseen degradations. For example, the pre-trained EDSR is trained on the domain of clean images, and our SRTTA is able to adapt the pre-trained EDSR model to the unseen domain such as Gaussian blur or JPEG during the test time, as shown in Table 1 of the paper. Furthermore, we have shown the visualization results on the real-world test images with unseen/unknown degradations, as shown in Figure 3 of the paper and Figure 5 of the supplementary materials.
>
> **Effectiveness on the unseen domain:** To further evaluate our SRTTA on unseen degradation, we evaluate our SRTTA on a new unseen domain with the degradation of processed camera sensor noise[2, 3], which is not used in the training phase of the SR model or that of the degradation classifier. In Table A, results show that our SRTTA method is also able to improve the model performance on this unseen degradation.
>
> **Effectiveness on the domains with mixed multiple degradations:** We also evaluate our SRTTA on the domain with mixed multiple degradations, where we change the degradation classifier to predict the probabilities of blur, noise, and JPEG degradation simultaneously. Experimental results in Table A of General Response 2 demonstrate the effectiveness of our SRTTA on these domains.
>
> Table A. Results of different methods on **unseen** domain with the degradation of processed camera sensor noise [2,3]. We report the PSNR and SSIM values of different methods on 100 images with random processed camera sensor noise.
>
> | Methods | PSNR | SSIM |
> |---|---|---|
> | SwinIR | 19.45 | 0.4960 |
> | IPT | 19.51 | 0.5005 |
> | HAT | 21.52 | 0.5957 |
> | DDNM | 19.63 | 0.5179 |
> | DAN | 21.53 | 0.5979 |
> | DCLS-SR | 21.57 | 0.6049 |
> | DualSR | 21.14 | 0.5861 |
> | MZSR | 20.40 | 0.4379 |
> | ZSSR | 21.57 | 0.6214 |
> | ZSSR+KernelGAN | 20.60 | 0.5430 |
> | EDSR | 21.56 | 0.6008 |
> | SRTTA (Ours) | **21.76** | **0.6503** |
>
> [1] DSLR-quality photos on mobile devices with deep convolutional networks, ICCV 2017.
> [2] Unprocessing images for learned raw denoising, CVPR 2019.
> [3] Designing a Practical Degradation Model for Deep Blind Image Super-Resolution, ICCV 2021.
>
>
> > Q3. Lack of evaluation results in terms of human eye-related metrics.
>
> A3: Our SRTTA is able to remove the degradation and generate images with better visual quality. To provide a more comprehensive assessment, we further evaluate different methods in terms of the **Fréchet Inception Distance (FID)** [1] and the **Learned Perceptual Image Patch Similarity (LPIPS)** distance [2], which correlate well with perceived image quality and are commonly used to evaluate the quality of generated images [3, 4]. Note that a lower FID means that the visual quality and distribution distance of the generated image is more similar to those of the ground-truth image. And a lower LPIPS value means the generated image is more perceptually similar to the ground-truth image.
>
> We evaluate different methods in terms of FID[1] and LPIPS[2] on two datasets, including **DIV2K-C** and **DIV2K-MC**. Note that **DIV2K-MC** contains four domains with a combination of multiple degradations, including BlurNoise, BlurJPEG, NoiseJPEG, and BlurNoiseJPEG. The domain "BlurNoiseJPEG" indicates that the test images in this domain contain the combined degradation of blur, noise and JPEG simultaneously. As shown in Table B, our SRTTA with a multi-label classifier (SRTTA-MC) achieves the lowest values of both FID and LPIPS scores, demonstrating our SRTTA is able to generate images with higher visual quality.
>
> Table B. Comparison results of our SRTTA with different methods in terms of FID[1] and LPIPS[2]. The reported FID/LPIPS values are the averages of all test images from the two datasets DIV2K-C and DIV2K-MC, respectively. Lower values indicate better performance.
>
> | Methods | DIV2K-C | DIV2K-MC |
> |:---:|:---:|:---:|
> | HAT | 64.92 / 0.2352 | 60.73 / 0.2640 |
> | SwinIR | 72.90 / 0.2441 | 60.62 / 0.2781 |
> | IPT | 68.22 / 0.2345 | 58.24 / 0.2453 |
> | KernalGAN+ZSSR | 88.28 / 0.2160 | 80.19 / 0.2371 |
> | MZSR | 68.27 / 0.2085 | 162.72 / 0.4463 |
> | DDNM | 70.80 / 0.2101 | 59.64 / 0.2083 |
> | DAN | 73.59 / 0.2260 | 56.96 / 0.2263 |
> | DCLS-SR | 83.44 /0.2472 | 57.93 / 0.2299 |
> | ZSSR | 56.66 / 0.1931 | 52.78 / 0.2152 |
> | EDSR | 69.70 / 0.2242 | 57.95 / 0.2338 |
> | SRTTA-MC(Ours) | 54.13 / **0.1927** | **37.71 / 0.1787** |
> | SRTTA-lifelong-MC(Ours) | **51.25** / 0.1964 | 45.68 / 0.1988 |
>
> [1] GANs Trained by a Two Time-Scale Update Rule Converge to a Local Nash Equilibrium, NeurIPS 2017.
> [2] The unreasonable effectiveness of deep features as a perceptual metric, CVPR 2018.
> [3] Improving diffusion models for inverse problems using manifold constraints, NeurIPS 2022.
> [4] Designing a practical degradation model for deep blind image super-resolution, ICCV 2021.

---

> > ### Comment · Reviewer_G7Pf · 2023-08-15
> > **the authors provide comprehensive responses**
> >
> > The authors provide comprehensive responses to the raised concerns, reaffirming the efficacy of their SRTTA method in handling real-world degradations, adapting to unseen degradation types, and producing images with improved visual quality.

---

> > > ### Author Response · Authors · 2023-08-16
> > > **Thanks to Reviewer G7Pf**
> > >
> > > Dear Reviewer G7Pf,
> > > We would like to thank you again for your time and efforts in reviewing our paper. Your valuable comments have helped to further strengthen our paper a lot!
> > >
> > > Sincerely,
> > > Authors of #2666

---

> ### Author Response · Authors · 2023-08-11
> **Looking forward to the response from Reviewer G7Pf**
>
> Dear Reviewer G7Pf,
> We have tried our best to respond to all of your comments. We really hope that our response helps to address all your concerns. Kindly let us know if you have any other concerns, and we will do our best to address them.
> Best regards,
> Authors of #2666

---

### Official Review · Reviewer_tgyp · 2023-07-05

**Soundness:** 3 good
**Presentation:** 3 good
**Contribution:** 2 fair
**Rating:** 5
**Confidence:** 4

**Summary:**

This paper proposed a fast test time adaption method for image SR. The authors point out that the existing image SR method use synthesized paired LR-HR data for training, which degradation type has a gap with the real-world sceneries. The authors also proposed a second-order based data synthesis circuit with a degradation classifier. The proposed method achieves impressive results on both real-world and synthesis data.

After reading the author's response, most of my problems have been addressed. Therefore, I would like to increase my initial score.

**Strengths:**

Strengths:
Experimental results show that the proposed method is effective for real-world image super-resolution.
Visual results are clear to show the improvement.

**Weaknesses:**

Weaknesses:
1. The concept of second-order degradations is already proposed by real-ESRGAN in 2021.
2. Please see the following detailed comments and suggestions.

**Questions:**

Here are some detailed comments and suggestions for improving the quality of this paper：
1.	In the abstract, I think it’s better to say ‘conventional SR’ or ‘PSNR/fidelity oriented SR’use paired data for training.
2.	Novelty problems, second order degradation is already proposed by real-ESRGAN, the authors need to enhance the explanation of the contribution of this paper.
3.	It is better to give a definition of the second-order degraded image like Equation.1
4.	The degradation category definition is a discrete single label, second order requires repetition and mixing, so is the design of the classifier reasonable? Did the authors consider the sequence of the degradations?
5.	The diagonal Fisher Information matrix in Section 4.3 requires independent identical distribution, is the image identically distributed under different degenerations? Equation 7 only reflects the stability of the same degradation under different augmentations, which does not seem to be able to target different degradations. In Equation 8, the parameter selection part I think is also not reasonable enough, through the artificial design of hyperparameters is usually difficult to meet the different datasets, so it is recommended to let the parameter be learnable.
6.	The visualization results look a bit over-smoothed, it is suggested to add some analysis to it.


**Limitations:**

See the above comments.

---

> ### Author Rebuttal · Authors · 2023-08-09
>
> > Q1. In the abstract, it's better to say 'conventional SR' or 'PSNR/fidelity oriented SR' use paired data for training.
>
> A1: We will revise the sentence of the abstract in the revised paper to avoid misleading.
>
> > Q2. Differences from Real-ESRGAN[1].
>
> A2: We would like to highlight the **difference** between our SRTTA with Real-ESRGAN in the following two folds:
>
> - **Solving different problems:**
> Real-ESRGAN tries to enumerate all the degradations in real-world scenes and train an SR model to solve the image restoration on any degradation. However, it is non-trivial to obtain all real-world degradations, **leading to domain shift issues when encountering unseen degradations** during testing, as shown in Figure 11 of real-ESRGAN [1].
> Unlike real-ESRGAN, our SRTTA aims to **adapt** the SR models to **different domains** when test images contain unknown/unseen degradations. Our second-order degradation scheme aims to construct the **pseudo paired data** to quickly update the SR model to the test domains.
>
> - **Different construction schemes:**
> Real-ESRGAN proposes the second-order degradation to construct the paired training data, whose low-resolution images are obtained from the ground-truth HR images. Then, the paired data is used to train an SR model in the **training phase** in a **supervised learning** manner, and the Real-ESRGAN model is fixed during the test time.
> Instead, our second-order degradation constructs the paired data using the test images with degradation (first-order degraded images). And our SRTTA model updates the SR model in a **self-supervised** manner to different domains during **testing**.
>
> Please refer to General Response 1 for a comprehensive explanation of the novelty and the contributions of our paper.
>
> [1] Real-ESRGAN: Training real-world blind super-resolution with pure synthetic data, ICCV 2021.
>
>
> > Q3. It is better to give a definition of the second-order degraded image like Equation.1
>
> A3: We will add a definition of the second-order degraded image in the revised paper.
>
> > Q4. The degradation category definition is a discrete single label, second order requires repetition and mixing, so is the design of the classifier reasonable? Did the authors consider the sequence of the degradations?
>
> A4: Our SRTTA is able to adapt to a wide range of degradation shifts. Indeed, the single-label classifier can limit the adaptation of SR models on the domains with mixed multiple degradations. To address this issue, we change the classifier to a multi-label classifier, which predicts the probabilities of multiple degradation types simultaneously. The results in Table A of General Response 2 demonstrate the effectiveness of our SRTTA with the multi-label classifier on the domains with mixed multiple degradations.
>
> When considering the sequence of the degradation in the second-order degradation process, it may contribute to more complex test domains, and we leave this investigation to our future work.
>
> > Q5. The diagonal Fisher Information matrix in Section 4.3 requires independent identical distribution, is the image identically distributed under different degenerations? Equation 7 only reflects the stability of the same degradation under different augmentations, which does not seem to be able to target different degradations.
>
> A5: Our SRTTA is able to preserve important parameters regardless of different degradation. The distribution of test images under different degradations may be different from each other. However, as explained in lines 211 to 212, we only use Equations (6) and (7) to compute the diagonal Fisher Information matrix once on the clean domain. **Regardless of different degradations**, we use the computed matrix to select important parameters to freeze during the whole test-time adaptation procedure.
>
> > Q6. In Equation 8, through the artificial design of hyperparameters is usually difficult to meet the different datasets, so it is recommended to let the parameter be learnable.
>
> A6: Our adaptive parameter preservation strategy only involves a single hyperparameter $\rho$ in Equation 8, which is not sensitive during the test-time adaptation. As shown in Table 4 of the paper, when $\rho$ is set within the range [0.2, 0.7], the results of our SRTTA-lifelong are slightly different, with fluctuations within 0.08 dB on the DIV2K-C test dataset. Consequently, we set $\rho$ to 0.5 for all datasets and domain data with different degradations.
>
> We agree that learning the parameter for each specific dataset can further improve the TTA performance, and we leave this to our future work.
>
> > Q7. The visualization results look a bit over-smoothed, it is suggested to add some analysis to it.
>
> A7: Our SRTTA is able to remove the degradation of real-world images and generate images with fewer artifacts, as shown in Figure 2 and Figure 3 of the paper. The results in Figure 2(a) look a bit over-smoothed, which may be caused by the improper removal of the original high-frequency information from the test image. Notably, denoising aims to eliminate the useless/noisy high-frequency information, which may inadvertently remove the original high-frequency information from sharp regions and lead to over-smoothed textures [1].
> Moreover, our second-order reconstruction loss is based on the Charbonnier penalty loss, which may result in generated images with over-smooth edges [2]. To compensate for the loss of high-frequency information, one can combine GAN loss [3] and perceptual loss [4] with our second-order reconstruction loss, following existing methods [3]. We leave this to our future work.
>
> [1] Accurate and fast image denoising via attention guided scaling, TIP, 2021.
> [2] Recovering realistic texture in image super-resolution by deep spatial feature transform. CVPR 2018.
> [3] Photo-realistic single image super-resolution using a generative adversarial network. CVPR 2017.
> [4] Perceptual losses for real-time style transfer and super-resolution, ECCV 2016.

---

> ### Author Response · Authors · 2023-08-11
> **Looking forward to the response from Reviewer tgyp**
>
> Dear Reviewer tgyp,
> We would like to thank you for your valuable comments on our paper. We sincerely hope that our response has addressed your initial concerns. If there are still unclear parts to you, please kindly let us know. We will do our best to address them.
> Best regards,
> Authors of #2666

---

> > ### Comment · Reviewer_tgyp · 2023-08-17
> >
> > After reading the author's response, most of my problems have been addressed. Therefore, I would like to increase my initial score to borderline accept.

---

> > > ### Author Response · Authors · 2023-08-17
> > > **Thanks to Reviewer tgyp**
> > >
> > > Dear Reviewer tgyp,
> > >
> > > We would like to thank you for increasing your score! Your valuable comments have helped to further strengthen the paper a lot!
> > >
> > > Sincerely,
> > > Authors of #2666

---

### Official Review · Reviewer_aSmb · 2023-07-05

**Soundness:** 3 good
**Presentation:** 3 good
**Contribution:** 3 good
**Rating:** 4
**Confidence:** 4

**Summary:**

The submission presents Test-time adaptation Super Resolution (SRTTA) via learning from the second-order degradation and reconstruction. This method super-resolve images with various degradation types while maintaining high efficiency. In addition, the paper also introduce a new dataset DIV2K-C, C stands for corruption, to better depict various image corruptions in the real-world SR use cases. The method is compared to other SOTA based this new dataset and achieves better metrics.

**Strengths:**

There are 2 major contributions in this submission:
1) Adaptive Parameter Preservation module for the test time adaptation;
2) DIV2k-C dataset.

**Weaknesses:**

I'm really doubt about the problem setting. First, modern ISP always removes noises like Poisson noise, most images are clean besides some compression artifacts. Second, as the authors admit, if an image has some corruptions, it usually has more than 1 type of corruptions. The submitted method based on a classifier to choose at most 1 types of corruption does not follow the real-world scenarios.  Both of these two weakness challenges the fundamental design of the method.
In addition, the method is only evaluated on the new dataset created by the authors, which is far from enough.

**Questions:**

Since your degradation classifier is to determine the types of the degradation, which is usually a local effect, what's the reason of putting the whole image into the classifier (I assume you also have to resize your input image to 224x224) instead of putting patches of your input image into the classifier?

**Limitations:**

Yes.

---

> ### Author Rebuttal · Authors · 2023-08-09
>
> > Q1. **Significance of the problem setting**: I'm really doubt about the problem setting. Modern ISP always removes noises like Poisson noise, most images are clean besides some compression artifacts.
>
> A1: In this paper, we seek to adapt a trained super-resolution (SR) model to new domains with unseen degradations. This problem is particularly practical and has gained great attention in the SR field [3], as the real-world environment may dynamically change and the collected images may continuously suffer from unseen degradations. We further clarify the degradation types considered in our SRTTA as follows:
>
> - **Noise degradations indeed commonly exist in the real-world collected images**. For example, traditional cameras may capture a noisy image with a high ISO value [1] or a short exposure time [2]. Moreover, ISP may introduce more complex degradation on the coloured images, which are unpredictable and varying among different devices [3].
>
> - **All degradation types considered in our SRTTA have been widely explored by prior SR methods [4,5]** or image recognition methods [6]. Here, our difference from prior SR methods is that the degradation scenarios we considered are more complex (total 8 degradations). In contrast, prior SR methods investigate a limited number of degradations (e.g., 1 type in [7] and 2 types in [8]). In this sense, our SRTTA is more practical for real-world applications.
>
> [1] Benchmarking denoising algorithms with real photographs, CVPR 2017.
> [2] Learning to see in the dark, CVPR 2018.
> [3] Blind image super-resolution: A survey and beyond. TPAMI 2022.
> [4] Designing a practical degradation model for deep blind image super-resolution, ICCV 2021.
> [5] Real-ESRGAN: Training real-world blind super-resolution with pure synthetic data, ICCV 2021.
> [6] Benchmarking Neural Network Robustness to Common Corruptions and Perturbations. ICLR 2019.
> [7] Unfolding the Alternating Optimization for Blind Super Resolution, NeurIPS 2020.
> [8] Learning a single convolutional super-resolution network for multiple degradations, CVPR 2018.
>
> > Q2. The SRTTA is inapplicable to domains with mixed multiple degradations.
>
> A2: Actually, our SRTTA framework can be applied to more complex real-world scenarios with mixed multiple degradations. In our experiments, we have investigated our SRTTA performance on 8 domains with single degradation. In the rebuttal, we further evaluate our SRTTA on four domains with mixed multiple degradations, including BlurNoise, BlurJPEG, NoiseJPEG and BlurNoiseJPEG. Please refer to General Response 2 for more details and results. Experimental results show that our SRTTA achieves an impressive improvement over existing methods in these domains, further demonstrating the effectiveness of our SRTTA.
>
>
> > Q3. The method is only evaluated on the new dataset created by the authors, which is far from enough.
>
> A3: We indeed have shown the experimental results on a real-world test dataset DPED[1] in Figure 3 of the paper and Figure 5 of the supplementary materials, which shows the effectiveness of our SRTTA in real-world scenarios. To further demonstrate the effectiveness of our method, we conduct experiments on more real-world datasets, including OST300[2] and ADE20K[3]. As shown in **Figure A of the supplementary PDF file accompanying this rebuttal**, experimental results demonstrate that our SRTTA is able to generate images with fewer artifacts for real-world test images.
>
> [1] DSLR-quality photos on mobile devices with deep convolutional networks, ICCV 2017.
> [2] Recovering realistic texture in image super-resolution by deep spatial feature transform, CVPR 2018.
> [3] Semantic understanding of scenes through the ADE20K dataset, IJCV 2019.
>
> > Q4. Since your degradation classifier is to determine the types of **the degradation, which is usually a local effect**, what's the reason of putting the whole image into the classifier (I assume you also have to **resize your input image** to 224x224) instead of putting patches of your input image into the classifier?
>
> A4: In our SRTTA, we input the whole image **without resize operation** (named **Whole Image**) into the classifier for degradation determination. However, as you suggested, we can also input the patches of the image into the degradation classifier (named **Patches**) to recognize the local effect. Here, we further compare these two types of inputs (**Whole Image** and **Patches**) in Table A. Results show that taking Patches as inputs achieves lower degradation classification accuracy, resulting inferior adaptation performance on the DIV2K-C dataset.
>
>
> Table A. Results of our SRTTA with different inputs of the degradation classifier. For Patches, we average the predicted results of all patches as the final predicted degradation type.
>
> | Input of the classifier | Accuracy | DIV2K-C  PSNR/SSIM | Total Inference Time  seconds/image |
> | :---: | :---: | :---: | :---: |
> | Whole Image | 0.9400 | 27.78/0.7290 | 5.38 |
> | Patches | 0.9122 | 27.64/0.7233 | 5.41 |

---

> > ### Comment · Reviewer_aSmb · 2023-08-15
> > **How did you input the whole image into the classifier?**
> >
> > Up to now, I still don't buy the point of inputting the whole image into your ResNet-50, especially, you train this classifier with only 224x224 patches, as stated in your supplemental material C.1.
> > The default input size of ResNet-50 is definitely 224x224, and your training makes this model also only learn the pixel relationships from that particular input size. It means the 7x7 and 3x3 kernels in this classifier are only familiar with pixel relationships in that resolution. How can it work on images with arbitrary resolutions, say 1080p, 1920x1080.

---

> > > ### Author Response · Authors · 2023-08-17
> > > **Response to the discussion on the details of the degradation classifier.**
> > >
> > > Dear Reviewer aSmb,
> > >
> > > Our classifier is able to work on images with arbitrary resolutions because we **do not change the pixel relationships** of images during both the training and the testing phase. We illustrate the reason as follows:
> > >
> > > - During the training phase, we **crop a 224 $\times$ 224 patch** from the training image **instead of resizing** the training image into 224 $\times$ 224. Here, the crop operation **does not change the pixel relationships** of the training image. In this case, the convolutional kernels in the classifier have the same receptive field on the original training image (e.g., 1920 × 1080) and the cropped patch (224 × 224). The reason why we crop training images into patches is to **increase the batch size on the limited GPU memory**. With a large batch size, the trained classifier tends to achieve better performance[1]. And the similar technique is also applied to the training of super-resolution (SR) models [2,3,4].
> > >
> > > - During testing, we directly input the whole test images into the classifier to recognize the degradation types, thus **the pixel relationships** of test images are also **unchanged**. Note that, equipped with an **adaptive global pooling** layer, the ResNet50 is able to deal with test images with arbitrary resolutions (larger than 224 $\times$ 224). Since the global pooling layer helps to aggregate the global information of the images during testing, the classifier with the whole image as input is able to achieve higher accuracy, as shown in Table A of Q4 (see the Table below).
> > >
> > > Table. Results of our SRTTA with different inputs of the degradation classifier. For Patches, we average the predicted results of all patches as the final predicted degradation type.
> > >
> > > | Input of the classifier | Accuracy | DIV2K-C  PSNR/SSIM | Total Inference Time  seconds/image |
> > > | :---: | :---: | :---: | :---: |
> > > | Whole Image (no resize) | 0.9400 | 27.78/0.7290 | 5.38 |
> > > | Patches | 0.9122 | 27.64/0.7233 | 5.41 |
> > >
> > > We sincerely hope our clarifications above have addressed your concerns. When making the final decision, we really hope that you could consider the novelty and contributions of our paper, which is illustrated in General Response 1. Kindly let us know if you have any further comments, and we are happy to continue the discussion.
> > >
> > > Best regards,
> > > Authors of #2666
> > >
> > > [1] Don't Decay the Learning Rate, Increase the Batch Size, ICLR 2018.
> > > [2] SwinIR: Image Restoration Using Swin Transformer, ICCV 2021.
> > > [3] Designing a Practical Degradation Model for Deep Blind Image Super-Resolution, ICCV 2021.
> > > [4] Real-ESRGAN: Training Real-World Blind Super-Resolution with Pure Synthetic Data, ICCV 2021.

---

> > > > ### Author Response · Authors · 2023-08-19
> > > > **Thanks to Reviewer aSmb**
> > > >
> > > > Dear Reviewer aSmb,
> > > >
> > > > We notice that you have raised your score, thank you very much. Kindly let us know if you have other questions, we are happy to continue the discussion.
> > > >
> > > > Best regards,
> > > > Authors of #2666

---

> ### Author Response · Authors · 2023-08-11
> **Looking forward to the response from Reviewer aSmb**
>
> Dear Reviewer aSmb,
> We have tried our best to address all the concerns and provided explanations to all questions. We sincerely hope that our answer has addressed your initial concerns. Kindly let us know if you have any other concerns, and we will do our best to address them.
> Best regards,
> Authors of #2666

---

### Official Review · Reviewer_pkZJ · 2023-07-07

**Soundness:** 3 good
**Presentation:** 3 good
**Contribution:** 3 good
**Rating:** 7
**Confidence:** 3

**Summary:**

The paper proposes to solve the domain shift problem for a pretrained super-resolution model. A pretrained classifier is utilized to predict the type of degradation of the test image. After the degradation type is detected, the degradation is used as an augmentation scheme for adaptation training for the new SR model.

**Strengths:**

The paper is easy to follow.

The motivation for the work is reasonable since super-resolution is an important task in Computer Vision, while the existing methods only focus on one degradation type. This results in the poor performance of the SR model in some cases.

**Weaknesses:**

The analysis is the weakest part of the work:

1. What are the consequences of domain shift when applying pretrained SR models? The work only mentions blur degradation is the most popular degradation type that other SR models concentrate on. However, no qualitative or quantitative observation is provided if another type of degradation happens in the dataset. What happens if the provided test datasets are degraded by blur and JPEG degradation? or both blur and noise degrade the provided test datasets? or only JPEG, or only noise?

2. In practice, which type of degradation happens the most? Whether the cover for the domain shift is necessary? Which scenarios will we need to do the domain adaptation training for the SR model?

3. The training of classifier in C.1 section mentions 8 types of degradation. What are the 8 types? Why, in the main paper, the classifier only has 4 classes?

4. The qualitative results in Figure 3 shows that KernelGAN+ZSSR is better than the proposed method.

5. In line 185, the author mentions, "encourage the pre-trained model to learn how to reconstruct ...", so the pretrained model will also be trained?

**Questions:**

see the weaknesses

(Weaknesses are addressed after rebuttal and discussions )

**Limitations:**

no limitations

---

> ### Author Rebuttal · Authors · 2023-08-08
>
> > Q1. What are the consequences of domain shift when applying pretrained SR models?
>
> A1: When the domain shift issue occurs, pre-trained SR models often generate HR images with unplausible artifacts, such as over-sharp edges or amplified noise [1, 2].
>
> [1] Blind super-resolution with iterative kernel correction, CVPR 2019.
> [2] Real-ESRGAN: Training real-world blind super-resolution with pure synthetic data, ICCV 2021.
>
> > Q2. Visualization of test images with different degradations.
>
> A2. Actually, we have provided visualizations of the test images from the DPED dataset, which contains a combined degradation of noise, blur and some unknown artifacts, as shown in Figure 3 of the paper and Figure 5 of the supplementary materials. We will provide more qualitative observations of test images with different degradations in the revised paper.
>
> > Q3. Evaluation on the test images with mixed multiple degradations.
>
> A3. We have investigated our SRTTA performance on 8 domains with single degradations, including blur, noise and JPEG degradations. In the rebuttal, we further evaluate our SRTTA on four domains with a combination of different degradation, including BlurNoise, BlurJPEG, NoiseJPEG and BlurNoiseJPEG. Experimental results show that our SRTTA significantly outperforms previous methods in these domains. Please refer to General Response 2 for more details and results.
>
> > Q4. Which type of degradation happens the most?
>
> A4: It is hard to know which type of degradation happens the most due to the innumerable images on the Internet. But most previous works [1,2] mainly focus on the common degradations of Gaussian Blur, Gaussian Noise, and JPEG compression.
>
> [1] Learning a single convolutional super-resolution network for multiple degradations, CVPR 2018.
> [2] Designing a Practical Degradation Model for Deep Blind Image Super-Resolution, ICCV 2021.
>
> > Q5. Whether the cover for the domain shift is necessary?
>
> A5: The domain shift poses a critical challenge in image super-resolution due to the diverse degradation types in real-world images [1]. Existing SR models trained on one domain (such as Gaussian Blur) tend to perform poorly on other common domains (such as Gaussian noise or JPEG). Consequently, the cover of the domain shift is necessary, allowing the adapted SR models to be applicable to diverse real-world scenes without requiring additional data collection or model retraining.
>
> [1] Blind image super-resolution: A survey and beyond, TPAMI 2022.
>
> > Q6. Which scenarios need to do the domain adaptation training for the SR model?
>
> A6: In real-world scenarios, the need for domain adaptation training (test-time adaptation) is common due to the diverse degradation types in real-world images. When the degradation of test images differs from that of training data, i.e., when the domain shift issue occurs, the SR models often require to perform test-time adaptation to adapt to these new domains. And the diverse degradation types are driven by several reasons.
>
> **First**, imaging devices differ greatly in the characteristics of the photos taken. Compared with a DSLR camera, a smartphone camera tends to produce "flattened" and noisy images [1].
> **Second**, due to the diversity of the imaging environment, real-world images, such as remote sensing images [2], may contain different types of degradation.
> **Moreover**, image processing algorithms, such as image compression, may also introduce complex degradation on real-world images [1].
>
> [1] Blind image super-resolution: A survey and beyond, TPAMI 2022.
> [2] Multilayer degradation representation-guided blind super-resolution for remote sensing images, TGRS, 2022.
>
> > Q7. The training of classifier in C.1 section mentions 8 types of degradation. What are the 8 types? Why, in the main paper, the classifier only has 4 classes?
>
> A7: The **8 degradation types** are Gaussian Blur, Defocus Blur, Glass Blur, Gaussian Noise, Poisson Noise, Impulse Noise, Speckle Noise and JPEG. Following the classical degradation model [1], we categorize these types into 3 distinct classes, namely **blur**, **noise** and **JPEG**. Notably, many images in real-world scenes are **clean** images, which require no need to perform test-time adaptation. Consequently, we filter out these images during the test-time adaptation. Finally, we set the classifier to predict the **4 classes**, including **clean**, **blur**, **noise**, and **JPEG**.
>
> **Motivation for predicting 4 classes:** In real-world scenes, test images may contain degradations other than these 8 types, such as ringing or overshoot artifacts [1], which can be viewed as specific variations of blur, noise or JPEG. Since it is infeasible to cover all the degradation types in real-world scenes, we make the degradation classifier to predict the **coarse-level** classes.
>
> [1] Real-ESRGAN: Training real-world blind super-resolution with pure synthetic data, ICCV 2021.
>
> > Q8. The results in Figure 3 show that KernelGAN+ZSSR is better than the proposed method.
>
> A8: The KernelGAN+ZSSR focuses on removing blur degradation from test images, leading to visually sharp results. But KernelGAN+ZSSR may amplify the noise in the test images, leading to generated images with increased artifacts in Figure 2 and Figure 3. Instead, our SRTTA identifies the noise degradation in the test image of Figure 3, resulting in images with significantly reduced noise levels. Moreover, the KernelGAN+ZSSR model is extremely slower than our SRTTA (231.41 vs. 5.38 seconds/image).
>
> > Q9. Will the pre-trained model also be trained?
>
> A9: During the test-time adaptation (TTA), we freeze all parameters of the pre-trained SR model, and we only update the TTA model (SRTTA). To avoid ambiguity, we will revise the paper accordingly.

---

> ### Author Response · Authors · 2023-08-11
> **Looking forward to the response from Reviewer pkZJ**
>
> Dear Reviewer pkZJ,
> We have addressed your initial concerns regarding our paper. We are happy to discuss them with you in the openreview system if you feel that there still are some concerns/questions. We also welcome new suggestions/comments from you!
> Best regards,
> Authors of #2666

---

> > ### Comment · Reviewer_pkZJ · 2023-08-20
> >
> > Dear authors,
> >
> > Thanks for the Rebuttal. I will increase my score to advocate the publication of the paper.
> >
> > However, I suggest the authors refine the following arguments:
> > > 1. "It is hard to know which type of degradation happens the most due to the innumerable images on the Internet. But most previous works [1,2] mainly focus on the common degradations of Gaussian Blur, Gaussian Noise, and JPEG compression."
> > >> It is best to have some statistics to back it up. At least based on some collected datasets from the internet or some famous benchmark datasets.
> > > 2. For Q5 and Q6, I suggest the authors revise the paper with more citations that the community suffers from the problem.
> > > 3. For Q1, I suggest the authors provide examples that visualize the problems when applying a pretrained model with Domain shift. The more details, the more interesting paper should be. For example, given a pretrained model with Gaussian Blur, what would happen when the test degradation is Poisson Noise or Impulse Noise? More examples would be appreciated.
> > > 4. For Q2, Please add more examples regarding each type in the 8 types of degradations mentioned by the authors. Many examples are encouraged.
> >
> > Best regards,

---

> > > ### Author Response · Authors · 2023-08-20
> > > **Response to Further Discussion**
> > >
> > > Dear Reviewer pkZJ,
> > >
> > > Thank you for your valuable comments. We provide a further response to the discussion as follows.
> > >
> > > > Q1. It is best to have some statistics to back it up. At least based on some collected datasets from the internet or some famous benchmark datasets.
> > >
> > > A1. Thank you for your suggestion. We count the statistics of the degradation types of real-world images and report the results in Table A. We use our degradation classifier to identify the degradation types of real-world images from five datasets, including RealSR [1], DRealSR [2], DPED [3], OST300 [4] and ADE20K [5]. Experimental results show that the degradation type of blur happens the most among these real-world datasets.
> > >
> > > Table A. The count of the predicted degradation types of the real-world images from the five datasets. Note that some images can contain more than one degradation type simultaneously.
> > > | Dataset (# Images) | Clean | Blur  | Noise | JPEG |
> > > |:----------------:|:-----:|:-----:|:-----:|:----:|
> > > |      RealSR (912)      |   77  |  804  |   22  |  113 |
> > > |      DRealSR (35148)   | 13052 | 22096 |   0   |   0  |
> > > |       DPED (187)       |   58  |   27  |   67  |  39  |
> > > |      OST300 (300)      |  163  |   29  |   22  |  95  |
> > > |      ADE20K (27574)    |  9815 |  3697 | 13333 | 2901 |
> > > |      Total (64121)     | 23165 | **26653** | 13444 | 3148 |
> > >
> > >
> > > [1] Toward real-world single image super-resolution: A new benchmark and a new model, ICCV 2019.
> > > [2] Component divide-and-conquer for real-world image super-resolution, ECCV 2020.
> > > [3] DSLR-quality photos on mobile devices with deep convolutional networks, ICCV 2017.
> > > [4] Recovering realistic texture in image super-resolution by deep spatial feature transform, CVPR 2018.
> > > [5] Semantic understanding of scenes through the ADE20K dataset. IJCV 2019.
> > >
> > >
> > > > Q2. For Q5 and Q6, I suggest the authors revise the paper with more citations that the community suffers from the problem.
> > >
> > > A2. Thanks for your suggestion. We will revise our paper to further include and discuss the relevant papers to show that the community suffers from the domain shift problem, such as [1,2,3,4].
> > >
> > > [1] Blind image super-resolution: A survey and beyond, TPAMI 2022.
> > > [2] Multilayer degradation representation-guided blind super-resolution for remote sensing images, TGRS, 2022.
> > > [3] Mutual-feed learning for super-resolution and object detection in degraded aerial imagery, TGRS, 2022.
> > > [4] Deep Blind Super-Resolution for Satellite Video, TGRS 2023.
> > >
> > >
> > > > Q3. I suggest the authors provide examples that visualize the problems when applying a pretrained model with domain shift.
> > >
> > > A3. Thank you for your valuable suggestion. In our paper, we have provided the visualization when applying a pre-trained model on different test domains. For example, we have evaluated the DAN model [1], which is **trained on the domain with Gaussian blur**, on the **test domains with Gaussian noise and JPEG compression**, as shown in Figure 2 of the paper and Figures 3-4 in the supplementary materials.
> > >
> > > To comprehensively investigate the effect of the domain shift problem, we will provide more visualization results when evaluating the pre-trained models on different test domains. For example, we will provide the visualization results when applying a pre-trained model, which is **trained on the domain with Gaussian noise**, on the **test domain with Gaussian blur or JPEG compression**. Moreover, we will also provide the visualization results when applying a pre-trained model, which is **trained on the domain with JPEG compression**, on the **test domain with Gaussian blur or Gaussian noise**.
> > >
> > > [1] Unfolding the alternating optimization for blind super resolution, NeurIPS 2020.
> > >
> > > > Q4. Please add more examples regarding each type in the 8 types of degradations mentioned by the authors.
> > >
> > > A4. Thanks for your suggestion. We will provide the visualization of some examples regarding each degradation type in the revised paper.
> > >
> > > We sincerely hope that our responses above have addressed your concerns. If you have other questions, we are happy to continue the discussion.
> > >
> > > Best regards,
> > > Authors of #2666

---

> > > > ### Author Response · Authors · 2023-08-21
> > > > **Thanks to Reviewer pkZJ**
> > > >
> > > > Dear Reviewer pkZJ,
> > > >
> > > > Thank you for increasing your score! Your invaluable comments have strengthened our paper a lot! We will include all the above suggestions in the revised paper.
> > > >
> > > > Best regards,
> > > > Authors of #2666

---

### Author Rebuttal · Authors · 2023-08-09

# **General Response**

> G1. Significance and novelty of our SRTTA.

In this paper, we seek to adapt a trained super-resolution (SR) model to domains with unseen degradations. This problem is practical and challenging, as the real-world environment may dynamically change and the collected images may continuously suffer from unseen degradations. To address this, we propose a novel test-time adaptation framework for SR models, which achieves an impressive improvement over existing methods with satisfying speed. We would like to highlight our **novelty** and **contributions** in the following three folds:

- **A novel test-time learning framework for SR model adaptation.** We propose a super-resolution test-time adaptation (SRTTA) framework to adapt any pre-trained SR models to different target domains during the test time. Without accessing any ground-truth high-resolution image, our SRTTA is applicable to practical scenarios with unknown degradation in a self-supervised manner.

- **A fast data construction scheme with second-order degradation.** We use a pre-trained classifier to identify the degradation type from test images and construct the paired data using the second-order degradation scheme. Since it is unnecessary to estimate the parameters of the degradation model, our second-order degradation scheme enables a rapid model adaptation to a wide range of degradation shifts.

- **A new dataset with eight different domains.** We construct a new test dataset named DIV2K-C, which contains eight common domains, to comprehensively evaluate the practicality of different SR methods. Previous methods mainly investigate the SR performance on a limited number of domains, while we generalize to more domains, which is more practical for real-world applications.

> G2. Good performance of our SRTTA on test images with mixed multiple degradations.

Our SRTTA can be definitely applied to super-resolve images with mixed multiple degradations and achieve excellent performance. To verify this, we extend the degradation classifier in SRTTA to a multi-label classifier, which simultaneously predicts the probabilities of blur, noise and JPEG degradation. Then, based on the predicted multiple degradations, we construct the second-order degraded image that contains multiple degradations for test-time model learning. We name this variant SRTTA-MC.

From the results in Table A, our SRTTA-MC achieves the best performance on 4 domains with different mixed degradations, e.g., 0.6187 (DualSR) $\rightarrow$ 0.7661 (our SRTTA-MC) regarding the average SSIM metric. These results further validate the effectiveness of our proposed methods.

Table A. Comparison with prior methods on **DIV2K-MC** dataset, which consists of four domains with the combination of multiple degradations. The term "BlurNoiseJPEG" indicates that the test images in this domain contain the combined degradation of blur, noise and JPEG simultaneously. We report the **PSNR($\uparrow$)/SSIM($\uparrow$)** values of different methods.

|     Methods |    BlurNoise | BlurJPEG | NoiseJPEG | BlurNoiseJPEG | Mean |
|:---:|:---:|:---:|:---:|:---:|:---:|
| HAT | 23.41/0.3992 | 28.86/0.7884 | 25.69/0.5721 | 24.42/0.5017 | 25.59/0.5654 |
| SwinIR | 20.91/0.3111 | 26.83/0.7481 | 23.86/0.5228 | 22.77/0.4502 | 23.59/0.5081 |
| IPT | 21.28/0.3273 | 26.83/0.7477 | 24.15/0.5346 | 22.96/0.4586 | 23.81/0.5171 |
| MZSR | 18.73/0.2129 | 24.90/0.6670 | 20.37/0.3977 | 20.62/0.3537 | 21.16/0.4078 |
| DualSR | 25.59/0.5613 | 28.24/0.7867 | 23.78/0.5855 | 24.62/0.5412 | 25.56/0.6187 |
| DAN | 24.14/0.4383 | 28.95/0.7907 | 26.20/0.5931 | 24.82/0.519 | 26.03/0.5853 |
| DCLS-SR | 23.84/0.4203 | 28.93/0.7905 | 26.37/0.5991 | 24.92/0.5233 | 26.02/0.5833 |
| DDNM | 22.62/0.3887 | 26.82/0.7462 | 25.11/0.5823 | 23.81/0.5041 | 24.59/0.5553 |
| ZSSR | 24.95/0.4933 | **29.02**/0.7934 | 26.68/0.6169 | 25.24/0.5421 | 26.47/0.6114 |
| ZSSR+KernelGAN | 23.08/0.4236 | 28.32/0.7864 | 21.90/0.4741 | 22.76/0.4427 | 24.02/0.5317 |
| EDSR | 24.02/0.4299 | 28.93/0.7903 | 26.08/0.5866 | 24.73/0.5136 | 25.94/0.5801 |
| TTA-C | 24.29/0.4462 | 28.93/0.7904 | 26.35/0.5975 | 24.91/0.5223 | 26.12/0.5891 |
| SRTTA-MC(Ours) | 26.82/0.6932 | 28.95/**0.7940** | 29.10/0.7878 | 26.97/0.7227 | 27.96/0.7494 |
| SRTTA-MC-lifelong(Ours) | **26.90**/**0.7346** | 29.00/0.7915 | **29.37**/**0.8014** | **27.36**/**0.7367** | **28.16**/**0.7661** |


> G3. Good performance of our SRTTA on practical scenarios.

Our SRTTA seeks to adapt a trained image SR model to domains with unknown degradations. In this sense, our SRTTA is particularly practical as the real world often dynamically changes and the collected image may suffer from various degradations. In our experiments, we have demonstrated the effectiveness of our SRTTA on a real-world dataset DPED[1] (see Figure 3 in the main paper and Figure 5 in the supplementary materials).

Here, we further conduct experiments on two more real-world datasets, i.e., OST300[2] and ADE20K[3], and put the visualization comparisons in **Figure A of the supplementary PDF file accompanying this rebuttal**. From the results, the generated images of our SRTTA contain less degradation, such as less noise on the flat area and textures with fewer ring artifacts. These results further demonstrate the effectiveness of our SRTTA on real-world collected images.

[1] DSLR-quality photos on mobile devices with deep convolutional networks, ICCV 2017.
[2] Recovering realistic texture in image super-resolution by deep spatial feature transform, CVPR 2018.
[3] Semantic understanding of scenes through the ADE20K dataset. IJCV 2019.

---

> ### Comment · Reviewer_G7Pf · 2023-08-15
>
> The authors present a compelling contribution in the field of super-resolution. The authors reiterate the novelty of their approach and the performance demonstrated through their experiments, and they have addressed all the raised concerns comprehensively. I am inclined to increase my score and recommend accepting this paper.

---

### Comment · Area_Chair_TPgr · 2023-08-15
**Reviewer-author discussion**

Many thanks to the reviewers who have provided feedback to the authors.

Dear @Reviewer tgyp, @Reviewer pkZJ, and @Reviewer aSmb,

We kindly request you to take a moment to review the authors' responses and share your opinions or any remaining concerns at your earliest convenience. Your input is greatly appreciated. Thank you.

AC

---

### Decision · Program_Chairs · 2023-09-21

**Decision:**

Accept (poster)

**Comment:**

The reviewers recognize the authors' contribution to the field of super-resolution. They acknowledge the authors' explanation of how their approach can handle images with arbitrary resolutions and the supportive experimental evidence provided. The paper's proposal of a fast test-time adaptation method for image super-resolution is noted, especially in the context of addressing the degradation shift issue between training and test images. The inclusion of a second-order degradation scheme and its integration with a degradation classifier is recognized as a significant advancement. The reviewers indicate that most of their concerns have been satisfactorily addressed and express intent to increase their initial score, recommending the paper's acceptance.